# ExSTED microscopy reveals contrasting functions of dopamine and somatostatin CSF-c neurons along the lamprey central canal

Elham Jalalvand[1], Jonatan Alvelid[1], Giovanna Coceano[1], Steven Edwards[1], Brita Robertson[2], Sten Grillner[2], Ilaria Testa[1]*

[1]Department of Applied Physics and Science for Life Laboratory, KTH Royal Institute of Technology, Stockholm, Sweden; [2]Department of Neuroscience, Karolinska Institutet, Stockholm, Sweden

**\*For correspondence:**
ilaria.testa@scilifelab.se

**Abstract** Cerebrospinal fluid-contacting (CSF-c) neurons line the central canal of the spinal cord and a subtype of CSF-c neurons expressing somatostatin, forms a homeostatic pH regulating system. Despite their importance, their intricate spatial organization is poorly understood. The function of another subtype of CSF-c neurons expressing dopamine is also investigated. Imaging methods with a high spatial resolution (5–10 nm) are used to resolve the synaptic and ciliary compartments of each individual cell in the spinal cord of the lamprey to elucidate their signalling pathways and to dissect the cellular organization. Here, light-sheet and expansion microscopy resolved the persistent ventral and lateral organization of dopamine- and somatostatin-expressing CSF-c neuronal subtypes. The density of somatostatin-containing dense-core vesicles, resolved by stimulated emission depletion microscopy, was shown to be markedly reduced upon each exposure to either alkaline or acidic pH and being part of a homeostatic response inhibiting movements. Their cilia symmetry was unravelled by stimulated emission depletion microscopy in expanded tissues as sensory with 9 + 0 microtubule duplets. The dopaminergic CSF-c neurons on the other hand have a motile cilium with the characteristic 9 + 2 duplets and are insensitive to pH changes. This novel experimental workflow elucidates the functional role of CSF-c neuron subtypes in situ paving the way for further spatial and functional cell-type classification.

## Editor's evaluation

Here, the authors use a variety of optical super-resolution techniques to explore the structure and function of different neurons in tissue. They present very interesting evidence that sensory neurons contacting the cerebrospinal fluid in lamprey differ in the motility of their cilium and in their response to variations of pH: while somatostatin-positive ciliated sensory neurons lose dense core vesicles from their soma and enrich them in their axons, dopaminergic ciliated sensory neurons do not show any change in DSV localization / density. This manuscript is of broad interest for the neuroscience and imaging community.

## Introduction

The vertebrate spinal cord contains neurons with many different functions. In the middle, there is the central canal containing cerebrospinal fluid. The wall of the central canal is lined with ciliated cerebrospinal fluid-contacting (CSF-c) cells in all vertebrates (*Agduhr, 1922*; *Vigh et al., 2004*). In the lamprey,

many CSF-c neurons are GABAergic sometimes with co-transmitters as somatostatin, neurotensin or dopamine (*Christenson et al., 1991*; *Dale et al., 1997*; *Jalalvand et al., 2014*; *Rodicio et al., 2008*). CSF-c neurons located at the lateral aspect of the central canal wall coexpress somatostatin and GABA and send axonal processes to mechanosensitive edge cells on the lateral margin of the spinal cord. GABA/somatostatin-expressing CSF-c neurons have recently been demonstrated to act as pH and mechanosensors and be part of a pH homeostatic system. At any deviation from neutral pH their activity is increased, which in turn leads to a depression of motor activity (*Jalalvand et al., 2016a*; *Jalalvand et al., 2016b*), which should counteract the processes that have led to the pH deviation. The accurate regulation of pH in the nervous system is of critical importance, and the fact that the GABA/somatostatin-expressing CSF-c neurons respond readily to pH changes gives them a particular homeostatic role. The effect of lowering the pH is blocked by APETx2, a specific acid-sensing ion channel 3 (ASIC3) antagonist (*Diochot et al., 2004*), as is their mechanosensitivity. The mechanism by which the GABA/somatostatin-expressing CSF-c neurons cause a reduction of locomotion has been suggested to be through release of somatostatin, since the effect on motor activity is blocked by somatostatin antagonists (*Jalalvand et al., 2016a*). CSF-c cells in a more ventral location contain dopamine (*Brodin et al., 1990*; *Schotland et al., 1996*) and their function is yet unknown.

In the present study, these two classes of CSF-c neurons, and their respective role for the operation of the pH homeostatic system and for the fluid transport in the central canal of the spinal cord is investigated. By applying a novel multi-scale imaging approach, using a high-throughput imaging method with sufficient resolution to resolve individual cells and organelles, we have been able to unravel the mode of operation of these two subtypes of CSF-c neurons. Light-sheet microscopy is the method of choice for rapid and minimally invasive acquisitions of large portions of tissues, but with a compromised spatial resolution in favour of recording speed and minimal photo-bleaching. To overcome this problem, we combine light-sheet with expansion microscopy (ExLSM) (*Düring et al., 2019*), a technique that physically expands tissue samples by a procedure including sample-embedding in a polyacrylamide gel and ends with ~four- to fivefold larger transparent samples (*Chen, 2015*; *Tillberg et al., 2016*). The physical expansion of the sample allows imaging of large volumes of spinal cord tissue with single-cell resolution and access to fine spatial information. This imaging approach allows quantification of the relative abundance and spatial patterns of CSF-c neuronal subtypes with specific focus on somatostatin- and dopamine-expressing cells along the central canal.

To further understand the physiological role of dopamine- and somatostatin-expressing CSF-c neurons, we take advantage of stimulated emission depletion (STED) microscopy to profile their neurotransmitter spatial distribution inside CSF-c neurons. STED microscopy (*Hell and Wichmann, 1994*; *Willig et al., 2006*) reaches a spatial resolution of ~40 nm, which allows us to resolve single synaptic vesicles even when the organelles are densely packed. This approach can identify neurotransmitter-specific release of dense-core synaptic vesicles during basal activity and upon pH stimulation in both somatostatin- and dopamine-expressing CSF-c neurons.

To gain structural information at ~5–10 nm level, an additional spatial resolution increase is needed, which we demonstrate with the application of STED imaging on expanded spinal cord tissues (ExSTED) (*Gao et al., 2018*). ExSTED imaging features an effective lateral spatial resolution of <10 nm, and allowed us to obtain cell-type-specific structural insight, previously accessible only with electron microscopy, on the cilia subtypes in ciliated somatostatin- and dopamine-expressing CSF-c. The cilia symmetry differs between primary (sensory) cilia (9 + 0 microtubule duplets) and motile cilia (9 + 2 microtubule duplets) and could be investigated in the context of specific cell types thanks to the spatial and specificity abilities of ExSTED. We could show that dopamine CSF-c neurons have motile cilia that may contribute to the flow of the cerebrospinal fluid, whereas somatostatin CSF-c neurons instead have predominantly sensory cilia conveying pH and mechanosensitivity.

Overall, this study uses recently developed high-resolution imaging techniques adapted to profiling the CSF-c neurons in the spinal cord. This provides cell-type information down to the molecular level. The experimental design developed in this study can be applied also to other types of tissue.

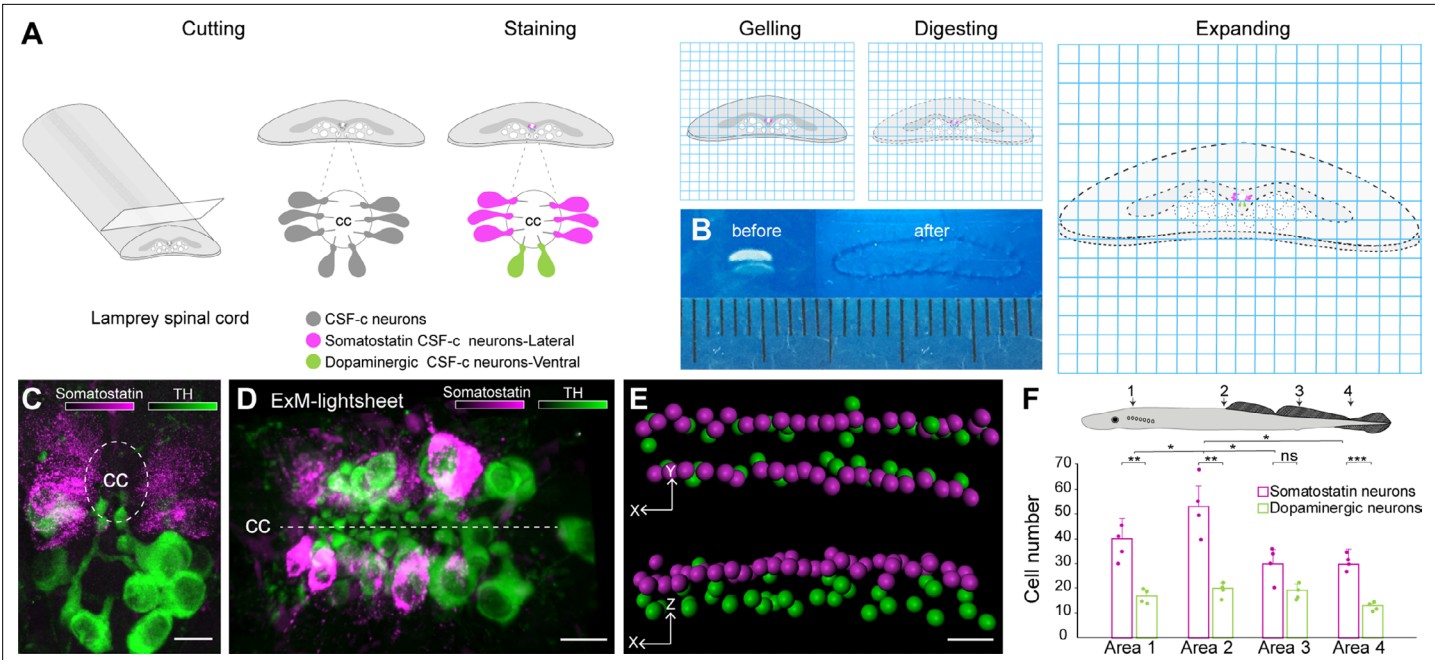

**Figure 1.** Somatostatin and dopaminergic cerebrospinal fluid-contacting (CSF-c) neurons distribution along the spinal cord by expansion and light-sheet microscopy. (**A**) A schematic illustration of the lamprey spinal cord treated for expansion microscopy (ExM). The spinal cords were immunostained for somatostatin and tyrosine hydroxylase (TH) prior to the ExM steps (MA-NHS treatment, gelation, proteinase K treatment, and expansion in water). (**B**) The spinal cord slices are shown before and after expansion. (**C, D**) Expanded samples imaged by light-sheet microscopy along the spinal cord. (**C**) Transverse and (**D**) horizontal images of somatostatin (magenta) and dopaminergic (green) CSF-c neurons shown by ExM-light-sheet microscopy. Scale bar, 30 μm. (**E**) Segmentation of the three-dimensional (3D) data from CSF-c neurons. Scale bar, 30 μm. (**F**) Quantification of somatostatin- and dopamine-expressing CSF-c neurons in four different areas of the spinal cord. The data are represented as the mean of number of cells in volume of each area; the error bar represents SD; Student's paired $t$-test: *$p < 0.05$ significant difference of somatostatin CSF-c neurons area 1 vs area 2 ($p = 0.016$, $t_3 = -4.84$), area 2 vs area 3 ($p = 0.016$, $t_3 = 5.72$) and vs area 4 ($p = 0.04$, $t_3 = 3.38$), **$p < 0.01$ and ***$p < 0.001$ significant difference of somatostatin and dopamine CSF-c neurons at area 1 ($p = 5.8 \times 10^{-3}$, $t_3 = 7.06$), at area 2 ($p = 4 \times 10^{-3}$, $t_3 = 7.67$), at area 4 ($p = 7.9 \times 10^{-4}$, $t_3 = 13.9$), and non-significant difference (n.s.) at area 3 ($p = 0.09$, $t_3 = 2.40$). cc, central canal.

The online version of this article includes the following video and source data for figure 1:

**Source data 1.** Distribution of somatostatin and dopaminergic cerebrospinal fluid-contacting (CSF-c) neurons along the spinal cord.

**Figure 1—video 1.** Three-dimensional (3D) light-sheet with expansion microscopy (ExLSM) explores spatial organization of somatostatin and dopaminergic cerebrospinal fluid-contacting (CSF-c) neurons in the spinal cord.

https://elifesciences.org/articles/73114/figures#fig1video1

## Results

### Distribution of somatostatin- and dopamine-expressing CSF-c neurons along the spinal cord

To investigate the spatial distribution of CSF-c neurons around the central canal at the single-cell level with high speed and sufficient spatial resolution, we combined expansion and light-sheet microscopy on fluorescently labelled CSF-c neurons. An experimental design for expanding and handling spinal cord tissue was developed, which includes slicing, fluorescent immunolabelling of somatostatin and dopaminergic CFS-c neurons in 80–100 μm thick spinal cord slices, and the expansion steps with a final expansion factor of about ~4.5 (**Figure 1A, B**).

The combination of our spinal cord expansion protocol and light-sheet microscopy (ExLSM) enables us to record a volume of $360 \times 250 \times 200$ μm³ containing a large population of CSF-c neurons where the individual cells can be recognized and counted (**Figure 1C, D**). Somatostatin and dopaminergic (TH-expressing) CSF-c neurons can be visualized along the spinal cord (**Figure 1C, D**) and their specific location in the three-dimensional (3D) architecture of the tissue can be visualized (**Figure 1—video 1**) and quantified in different views (**Figure 1E**). The cell bodies of both dopamine and somatostatin CSF-c neurons are clearly visible, as well as their characteristic protrusion into the central

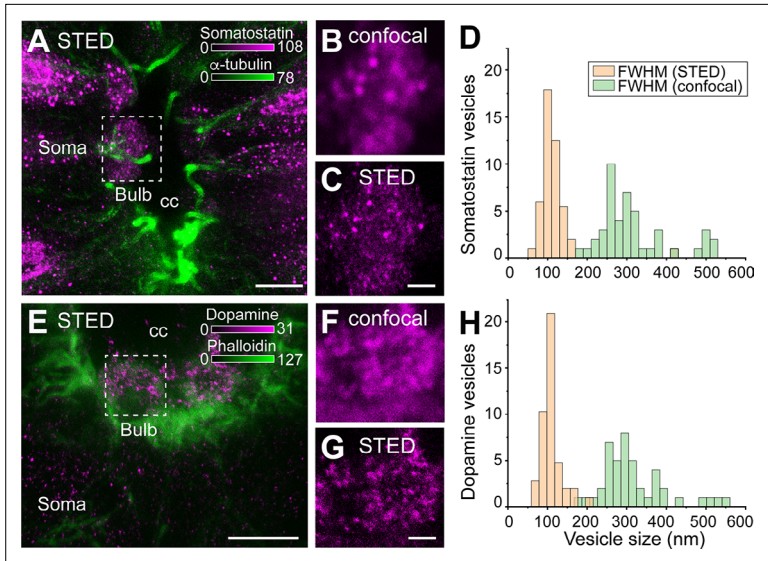

**Figure 2.** Somatostatin and dopamine in cerebrospinal fluid-contacting (CSF-c) neurons are stored in dense-core vesicles. (**A–C**) Somatostatin (magenta) and α-tubulin (green) immunostaining in CSF-c neurons. Scale bar in A, 5 µm. (**B, C**) Selected Region of interest (ROIs) of somatostatin dense-core vesicles (DCVs) in the bulb of somatostatin-expressing CSF-c neurons imaged with confocal and stimulated emission depletion (STED) microscopy, respectively. Scale bar, 500 nm. (**D**) Analysis of the size of somatostatin DCVs measured with confocal and STED microscopy (*n* = 46). (**E–G**) Dopamine (magenta) and phalloidin (green) immunostaining in CSF-c neurons. Scale bar in (**E**), 5 µm. (**F, G**) Selected ROIs of dopamine DCVs in the bulb of dopamine CSF-c neurons with confocal and STED microscopy, respectively. Scale bar, 500 nm. (**H**) Analysis of the size of dopamine DCVs with confocal and STED (*n* = 44). cc, central canal.

The online version of this article includes the following source data for figure 2:

**Source data 1.** Somatostatin in cerebrospinal fluid-contacting (CSF-c) neurons store in dense-core vesicles (DCVs).

**Source data 2.** Dopamine in cerebrospinal fluid-contacting (CSF-c) neurons store in dense-core vesicles (DCVs).

canal. Different types of CSF-c neurons show a specific distribution along the lamprey spinal cord in all three spatial dimensions. Somatostatin-expressing CSF-c neurons are found throughout the whole volume and located laterally relative to the central canal, while the dopaminergic CSF-c neurons are located ventrally, confirming that the observation on individual sections is maintained in large volume with a high degree of order.

The throughput of ExLSM allowed recording a total volume of $7.2 \times 10^7$ µm$^3$ along the spinal cord for a total of 224 cells. We quantified the distribution of CSF-c neuronal subtypes at four different levels of the spinal cord, resulting in a higher amount of somatostatin CSF-c neurons compared to the dopaminergic CSF-c neurons in all four areas of the spinal cord. Additionally, in a specific position (area 2) the somatostatin CSF-c neurons were more abundant than in other parts (*Figure 1F*).

## Somatostatin and dopamine neurotransmitters in CSF-c neurons are stored in dense-core vesicles

To investigate the subcellular location and compartmentalization of somatostatin and dopamine in CSF-c neurons, we used STED microscopy. The STED microscope was equipped with a glycerol objective and red-shifted wavelengths to allow tissue imaging with minimal spherical aberration and scattering.

Somatostatin and dopamine are stored in dense-core vesicles (DCVs) and were found in the soma, bulb protrusion (*Figure 2A–C,E–G*), and projections (data not shown) of the CSF-c neurons. No major differences between the somatostatin- and dopamine-positive puncta are observed. In both cases, the average diameter of somatostatin and dopamine DCVs was 100–120 nm with vesicles as small as 60 nm (full width at half maximum, FWHM; *Figure 2D, H*) measured with STED. However, using confocal microscopy, resulted in an overestimation of vesicle size with 250–300 nm (FWHM;

*Figure 2D, H*). Our data on the somatostatin and dopamine DCVs diameter recorded with STED are compatible with previously published electron microscopy data (*Schotland et al., 1996*).

## Somatostatin but not GABA release are responsible for pH response in somatostatin/GABA CSF-c neurons

The somatostatin CSF-c neurons in the lamprey spinal cord also express GABA (*Brodin et al., 1990*; *Christenson et al., 1991*; *Jalalvand et al., 2014*). We have recently shown that somatostatin/GABA CSF-c neurons are sensitive to pH changes of the cerebrospinal fluid (*Jalalvand et al., 2016a*; *Jalalvand et al., 2016b*). Here, we investigate how the spatial distribution and abundance of somatostatin and GABA changes during induced pH changes to acidic (6.5) or alkaline (8.5) conditions and if they were co-released. The somatostatin DCVs were visualized in the soma as well as in the axons by both confocal and STED microscopy (*Figure 3A–F*). Somatostatin DCVs were imaged with STED microscopy in control, acidic, and alkaline pH conditions. The number density of somatostatin DCVs measured for the soma of CSF-c neurons both in areas (*Figure 3G*) and volumes (*Figure 3—figure supplement 1*) decreased markedly in CSF-c neurons in either acidic or alkaline pH conditions. In contrast, the slices stained with a GABA antibody did not show any changes in fluorescence intensity at different pH (*Figure 3H–K*).

As somatostatin and GABA are co-expressed in the same CSF-c neurons, we investigated if they were colocalized in the same vesicles (*Figure 3L–P*). STED images of somatostatin DCVs and GABA-expressing CSF-c neurons did not show colocalization of somatostatin DCVs and GABA in the soma (*Figure 3M–O*). The GABA signal was not significantly different when measured in or outside of somatostatin DCVs, neither in the soma nor in the axons (*Figure 3P*). The results confirmed that there is no correlation between GABA and somatostatin signals in somatostatin vesicles. Our imaging data support the conclusion that during pH changes of the extracellular solution vesicles containing somatostatin are released, resulting in fewer DCVs in the soma, while GABA is not co-released.

## Dopaminergic CSF-c neurons are not sensitive to changes in extracellular pH

The next step was to explore whether the dopaminergic CSF-c neurons at the ventral aspect of the central canal were also sensitive to pH changes. As for the somatostatin experiments, acidic and alkaline extracellular pH was perfused on spinal cord slices and subsequently stained with a dopamine antibody to investigate the dopamine DCVs spatial distribution in dopaminergic CSF-c neurons. We performed STED microscopy to resolve and count the single vesicles in distinct neuronal locations (*Figure 4A–C*). The result showed no significant change in the number density of dopaminergic DCVs after perfusion with acidic or alkaline pH solutions, neither in cell areas (*Figure 4D*) nor in cell volumes (*Figure 4—figure supplement 1*).

To complement the results from the STED imaging, dopaminergic CSF-c neurons were patched as previously described for somatostatin-expressing CSF-c neurons (*Jalalvand et al., 2016a*; *Jalalvand et al., 2016b*). Their electrophysiological properties and response to changes of extracellular pH were investigated with whole-cell patch recording in current-clamp mode (*Figure 4E*). All recorded dopaminergic CSF-c neurons fired spontaneous action potentials, depolarizing synaptic potentials, and hyperpolarizing synaptic potentials (*Figure 4E*). To verify that responses were not evoked synaptically, the GABAergic and glutamatergic synaptic transmission was blocked by bath-application of gabazine (GABA$_A$ receptor antagonist) and kynurenic acid (glutamate receptor antagonist). During bath-applied extracellular solutions of acidic pH (6.5) or alkaline pH (8.5) no changes in spike frequency were observed, nor a net depolarization of the resting membrane potential (*Figure 4E1*). Thus, in contrast to the somatostatin CSF-c neurons the dopamine CSF-c are not sensitive to pH changes. All neurons labelled with Neurobiotin during the patch-clamp recording were tyrosine hydroxylase (TH) immunoreactive (*Figure 4F, H*).

Both somatostatin and dopamine CSF-c neurons are ciliated (see below) and the former known to be mechanosensitive (*Jalalvand et al., 2016a*). To test if the dopaminergic CSF-c neurons are mechanosensitive very brief fluid pulses were applied near their bulb protrusion in the central canal as previously done for the somatostatin CSF-c neurons (*Figure 4—figure supplement 2A*; *Jalalvand et al., 2016b*). Dopamine CSF-c neurons responded with an action potential and a distinct mechanosensitive response (*Figure 4—figure supplement 2B*). To investigate if the mechanosensitivity in

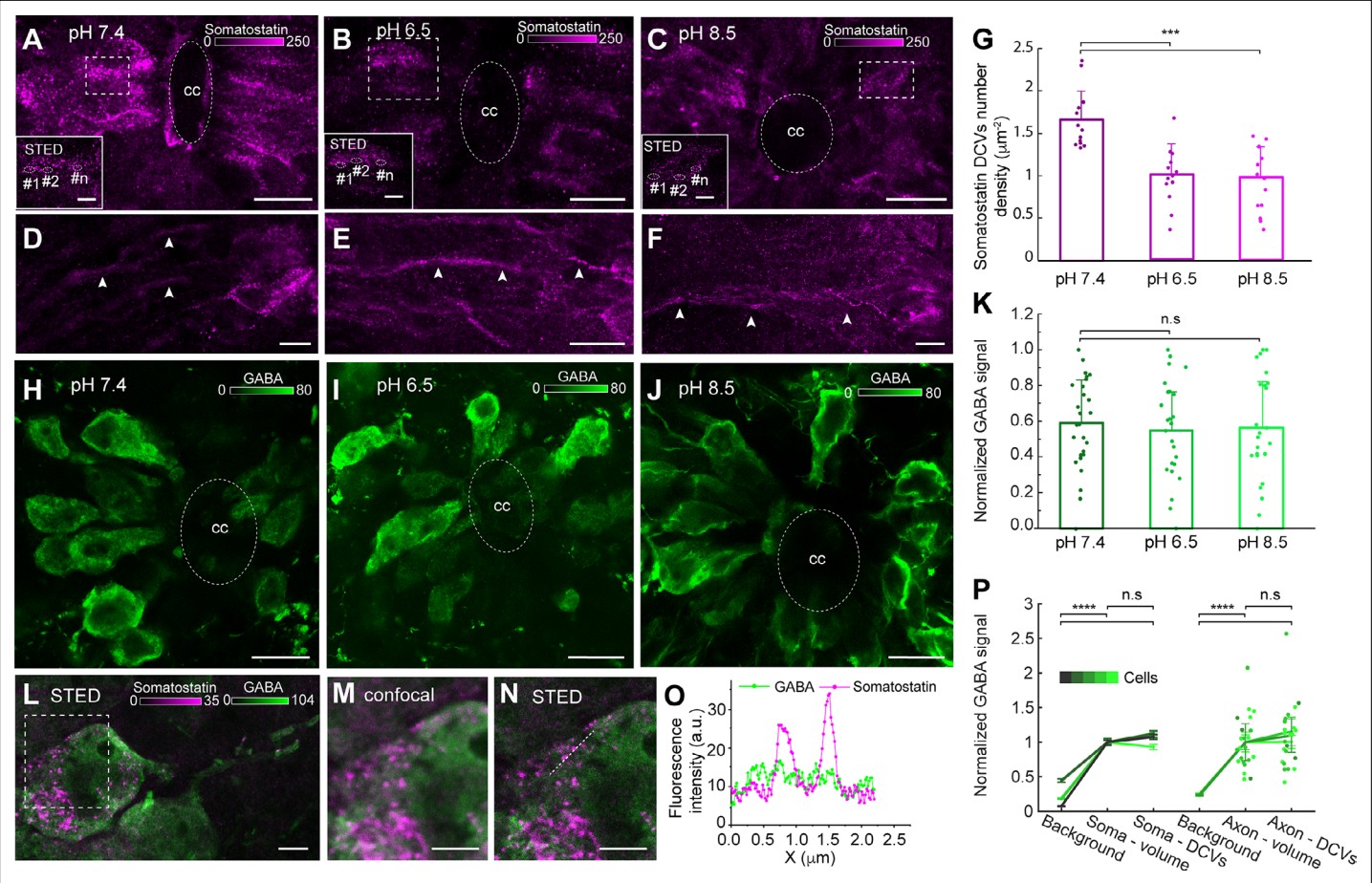

**Figure 3.** Acidic and alkaline pH decreased the number of somatostatin dense-core vesicles (DCVs) in the soma but did not affect gamma-Aminobutyric acid (GABA) intensity. (**A–F**) Spinal cord slices in normal (pH 7.4), acidic (pH 6.5), and alkaline (pH 8.5) extracellular solution stained with an anti-somatostatin antibody (magenta). (**A–C**) Confocal and stimulated emission depletion (STED) images (selected ROIs) of somatostatin DCVs in the soma. Scale bar in (A–C), 10 µm; in ROIs, 1 µm. (**D–F**) The axons of the somatostatin-expressing cerebrospinal fluid-contacting (CSF-c) neurons (arrowheads). Scale bar, 10 µm. (**G**) Quantification of somatostatin DCVs number density in cell area (µm$^{-2}$) in the different conditions ($n$ = 13). Student's paired $t$-test: ***$p < 0.001$ significant difference between pH 7.4 and 6.5 ($p = 5.26 \times 10^{-5}$, $t_{12} = 4.9$), and 7.4 and 8.5 ($p = 1.79 \times 10^{-5}$, $t_{12} = 5.3$). (**H–J**) The spinal cord slices in normal, acidic and alkaline extracellular solution, stained with an anti-GABA antibody (green). Scale bar, 10 µm. (**K**) Comparison of normalized GABA signals at pH 7.4 ($n$ = 26), 6.5 ($n$ = 24), and 8.5 ($n$ = 22), respectively. Student's $t$-test: non-significant difference (n.s.) between pH 7.4 and 6.5 ($p = 0.62$, $t_{47} = 0.48$), and 7.4 and 8.5 ($p = 0.80$, $t_{43} = 0.25$). (**L–O**) STED and confocal images of spinal cord slices stained for somatostatin (magenta) and GABA (green). (**L**) STED image of a CSF-c neuron. Scale bar, 1 µm. (**M, N**) Selected ROI from the soma in (I) shown at higher magnification with confocal and STED microscopy, respectively. Scale bar, 1 µm. (**O**) Line profile graph in image N. (**P**) Mean GABA signal in cellular compartments and compared to extracellular background ($n$ = 5), normalized to volume intensity in soma ($n$ = 5) and axons ($n$ = 3), respectively. Repetitions are different cells. Student's $t$-test between means of cellular means: ****$p < 0.0001$ significant difference between soma-volume and background ($p = 1.0 \times 10^{-5}$, $t_8 = -9.7$), soma-DCVs and background ($p = 1.0 \times 10^{-5}$, $t_8 = -9.6$), axon-volume and background ($p = 8.0 \times 10^{-8}$, $t_4 = -93$), and axon-DCVs and background ($p = 4.4 \times 10^{-5}$, $t_4 = -19$), non-significant differences (n.s.) between soma-volume and DCVs ($p = 0.090$, $t_8 = -1.9$), and axon-volume and DCVs ($p = 0.12$, $t_4 = 1.9$). Data (**G, K, P**) are represented as means, with error bars representing standard deviation (SD) (**G, K**) or standard error of the mean (SEM) (**P**). cc, central canal.

The online version of this article includes the following source data and figure supplement(s) for figure 3:

**Source data 1.** Effect of acidic or alkaline pH on somatostatin dense-core vesicles (DCVs) number density in cell area of somatostatin-expressing cerebrospinal fluid-contacting (CSF-c) neurons.

**Source data 2.** Effect of acidic or alkaline pH on GABA signal in somatostatin-expressing cerebrospinal fluid-contacting (CSF-c) neurons.

**Source data 3.** No correlation between GABA and somatostatin signals in somatostatin-expressing cerebrospinal fluid-contacting (CSF-c) neurons.

**Figure supplement 1.** Effect of acidic or alkaline pH on somatostatin dense-core vesicles (DCVs) number density in cell volume of somatostatin-expressing cerebrospinal fluid-contacting (CSF-c) neurons.

**Figure supplement 1—source data 1.** Effect of acidic or alkaline pH on somatostatin dense-core vesicles (DCVs) number density in cell volume of somatostatin-expressing cerebrospinal fluid-contacting (CSF-c) neurons.

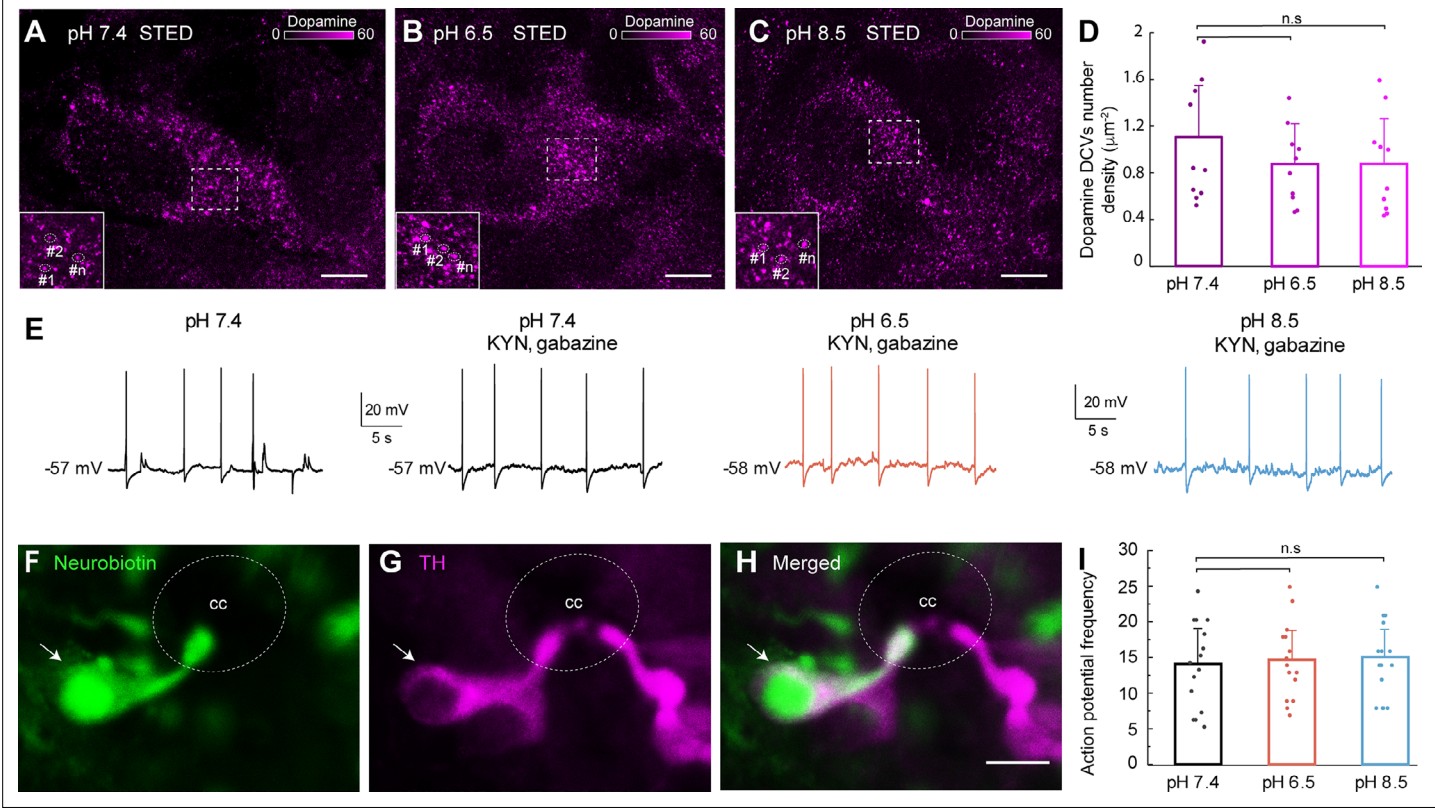

**Figure 4.** Dopaminergic cerebrospinal fluid-contacting (CSF-c) neurons did not respond to acidic and alkaline pH. (**A–C**) Stimulated emission depletion (STED) images of dopamine-containing dense-core vesicles (DCVs) in the soma of CSF-c neurons in normal (pH 7.4), acidic (pH 6.5), and alkaline (pH 8.5) extracellular solution. Scale bar, 1 μm. (**D**) Quantification of the number of dopamine DCVs number density in cell area (μm$^{-2}$) in the different conditions ($n$ = 10). Student's $t$-test: non-significant (n.s.) between pH 7.4 and 6.5 (p = 0.27, $t_9$ = 1.12), and 7.4 and 8.5 (p = 0.29, $t_9$ = 1.08). (**E**) Whole-cell patch recording of a CSF-c neuron, showing firing spontaneous action potentials in control (pH 7.4), acidic (p H 6.5), and alkaline (pH 8.5) conditions in the presence of gabazine (20 mM) and kynurenic acid (2 mM). (**F–H**) Photomicrographs of the CSF-c neurons recorded in (**E**) intracellularly filled with Neurobiotin (arrow) during recording. The labelled cell showed immunoreactivity to tyrosine hydroxylase (TH, arrow). Scale bar, 10 μm. (**I**) Action potential frequency during 1 min in CSF-c neurons at pH 7.4, 6.5, and 6.8, respectively ($n$ = 15). Student's paired $t$-test: non-significant difference (n.s.) between pH 7.4 and 6.5 (p = 0.24, $t_{14}$ = −1.22), and 7.4 and 8.5 (p = 0.1, $t_{14}$ = −1.75). The bar graph data are represented as the means, with error bars representing standard deviation (SD). cc, central canal.

The online version of this article includes the following source data and figure supplement(s) for figure 4:

**Source data 1.** Effect of acidic or alkaline pH on dopamine dense-core vesicles (DCVs) number density in cell area of dopaminergic cerebrospinal fluid-contacting (CSF-c) neurons.

**Source data 2.** Effect of acidic and alkaline pH on action potential frequency in dopaminergic cerebrospinal fluid-contacting (CSF-c) neurons.

**Figure supplement 1.** Effect of acidic or alkaline pH on dopamine dense-core vesicles (DCVs) number density in cell volume of dopaminergic cerebrospinal fluid-contacting (CSF-c) neurons.

**Figure supplement 1—source data 1.** Effect of acidic or alkaline pH on dopamine dense-core vesicles (DCVs) number density in the cell volume of dopaminergic cerebrospinal fluid-contacting (CSF-c) neurons.

**Figure supplement 2.** Dopaminergic cerebrospinal fluid-contacting (CSF-c) neurons are sensitive to fluid movement.

dopaminergic CSF-c neuron was mediated by the ASIC3 as in somatostatin CSF-c neurons, APETx2, a specific blocker of ASIC3, was applied. In the presence of APETx2 we still observed a mechanosensitive response (*Figure 4—figure supplement 2C*). Thus, the mechanosensitivity is not mediated by ASIC3 in dopamine CSF-c neurons. In situ hybridization showed expression of polycystic kidney disease 2-like 1 (PKD2L1) channels in dopaminergic CSF-c neurons (*Figure 4—figure supplement 2D*). As PKD2L1 has been confirmed as a mechanosensitive ion channel in zebrafish (*Böhm et al., 2016*; *Sternberg et al., 2018*), the results suggest that the mechanosensitivity of the dopaminergic CSF-c neurons may be mediated by PKD2L1 channels. In contrast, the mechanical response of somatostatin CSF-c neurons is blocked by an ASIC3 antagonist.

In conclusion, dopamine CSF-c neurons are not sensing pH changes as their somatostatin counter-parts, but both are mechanosensitive. The mechanical transduction is blocked by an ASIC3 antagonist in somatostatin CSF-c neurons and may be mediated by PKD2L1 in the dopamine CSF-c neurons as in the zebrafish.

## CSF-c neurons show both primary and motile cilia symmetries

Somatostatin- and dopamine-expressing CSF-c neurons are thus both mechanosensitive, but through different transduction mechanisms. Since both types of CSF-c neurons are ciliated, we investigated whether the functional difference is reflected in the structural organization of the cilia.

Cilia express acetylated α-tubulin and can be classified either as primary cilia, mainly present in sensory cells and neurons and with a 9 + 0 symmetry of nine outer microtubule doublets, or motile cilia, with a 9 + 2 symmetry showing an extra pair of microtubule singlets in the centre (*Gaertig and Wloga, 2008*; *Satir, 2005*). Primary and motile cilia both have an average diameter of 200–240 nm, close to the achievable spatial resolution of a confocal microscope equipped with a high numerical aperture objective. Therefore, a higher spatial resolution is crucial to assess their organization and subcompartmentalization.

One aim for exploring the cilia symmetry of ciliated CSF-c neurons was to uncover potential signalling compartments related to pH and mechanosensitivity of the CSF-c neurons. Cilia in the central canal was immunolabelled for acetylated α-tubulin, a protein characteristic of cilia, and imaged with confocal and STED microscopy to measure their diameters, respectively (*Figure 5A, B, E, F*). STED imaging visualizes the cilium as a hollow structure with an outer diameter of about 240 nm thanks to the increased spatial resolution, but the resolution is still not enough to detect the microtubule doublets. One strategy to increase the observable level of detail is to expand the tissue. In the expanded spinal cord tissue, pre-immunostained for acetylated α-tubulin, the resolution of the dense cilia in the central canal increased (*Figure 5—figure supplement 1*, 3D XYZ image) as compared to confocal imaging of the non-expanded sample. Confocal imaging of the expanded sample shows spatial details compara-ble to STED images (*Figure 5C*) of the non-expanded one, therefore still not having the resolution (5–10 nm) needed to dissect the internal cilia structure and identify microtubule doublets. That level of detail is crucial to further investigate the nature of cilia as sensory or motile for each specific cell type.

By using the combination of STED microscopy and expanded tissue (ExSTED), we were able to add a factor of ~four to five to the typical 50 nm resolution of STED microscopy and therefore resolving at the smaller spatial scale of 5–10 nm (*Figure 5—figure supplement 2*). In this way, it was possible to separate high-density cilia within the 3D geometry of the central canal (*Figure 5—video 1*) due to the increased imaging resolution and further resolve their internal structure and allowing to classify them as motile or sensory (*Figure 5D*). In some cilia, the central pair of tubules was clearly observed with a peak-to-peak distance of 70 nm (*Figure 5P–R*).

Both types of cilia symmetries in the lamprey spinal cord were explored: sensory (*Figure 5I–M*) and motile cilia symmetries (*Figure 5N–R*). The presence of motile cilia in the central canal of the lamprey has been shown with electron microscopy (*Schotland et al., 1996*; *Vigh et al., 2004*), but with ExSTED we could detect both motile and sensory cilia symmetries in this area. As all CSF-c neurons, including somatostatin and dopaminergic CSF-c neurons in the lamprey spinal cord, are ciliated, the next step is to know which cilia type is related to which neuronal phenotypes and if they contribute to a specific neuronal function.

## Somatostatin CSF-c neurons have both primary and motile cilia, but dopaminergic CSF-c neurons have only motile cilia

To further investigate whether somatostatin-expressing CSF-c neurons (pH-sensitive) and dopaminergic CSF-c neurons (non-pH-sensitive) have different types of cilia, expansion was combined with STED micros-copy. Using dual-colour imaging of spinal cord slices, either somatostatin or dopamine (TH immunos-taining), together with α-tubulin immunostaining (*Figure 6*) was analysed. To have better access to the cilia and CSF-c neurons the central canal of the expanded spinal cord has been cut horizontally and flipped 90° with respect to the coverslip. The central canal can then be visualized in a cylindrical form (*Figure 6A–C*). To confirm what type of CSF-c neurons the cilia belong to, z-stack scanning for all the imaging was applied. In our data, 70 cilia belong to somatostatin-expressing CSF-c neurons (*n* = 70). Of these, 60 cilia were primary (9 + 0 symmetry) and 10 cilia were motile (9 + 2 symmetry) (*Figure 6D–K, T*). Additionally, 20

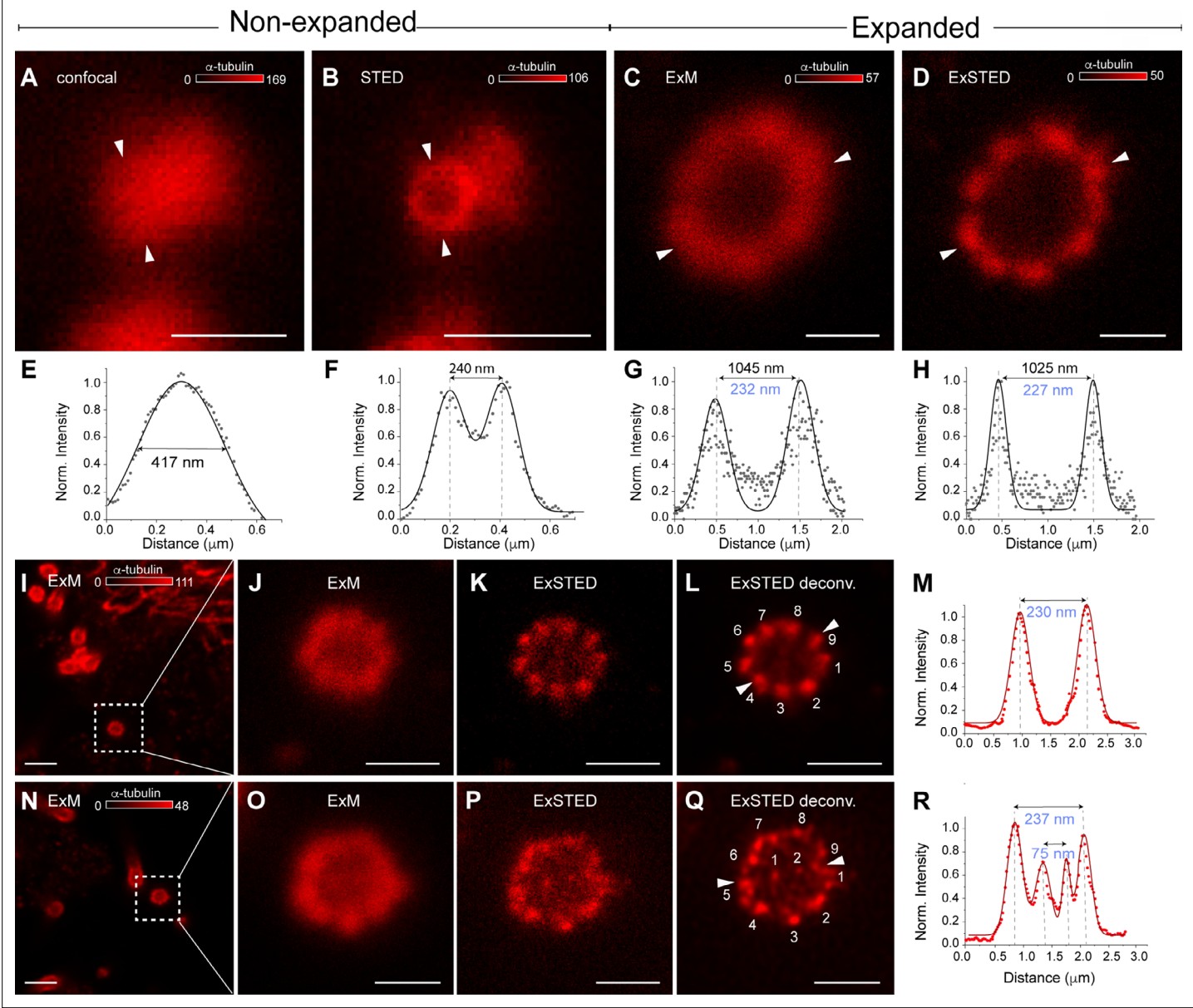

**Figure 5.** Primary and motile cilia symmetry are present in the lamprey spinal cord. (**A, B**) Confocal and stimulated emission depletion (STED) images of a cerebrospinal fluid-contacting (CSF-c) neuron cilium in a non-expanded spinal cord pre-stained with anti-α-tubulin antibodies. Scale bar, 0.5 μm. (**C, D**) Confocal (expansion microscopy, ExM) and STED (ExSTED) images of a cilium in the expanded spinal cord. Scale bar, 0.5 μm. (**E, F**) Quantification of the cilium diameter (arrowheads) in confocal and STED images in a non-expanded spinal cord. (**G, H**) Quantification of the cilium diameter (arrowheads) in confocal (ExM) and STED (ExSTED) in the expanded spinal cord. (**I–M**) Confocal (ExM, **I, J**) and STED (ExSTED, **K, L**) deconvoluted images of a primary cilium with 9 + 0 symmetry in the expanded spinal cord. Scale bar, I, 2 μm and J–L, 1 μm. (**M**) Quantification of the cilium diameter from image (**I**) (arrowheads). (**N–R**) Confocal (ExM, **N, O**) and STED (ExSTED, **P, Q**) deconvoluted images of a motile cilium with 9 + 2 symmetry in the expanded spinal cord. Scale bar, N, 2 μm and O–Q, 1 μm. (**R**) Quantification of the cilium diameter from image (**Q**) (arrowheads). Cilia diameters in blue were divided by the expansion factor.

The online version of this article includes the following video, source data, and figure supplement(s) for figure 5:

**Source data 1.** Quantification of a cilium diameter of cerebrospinal fluid-contacting (CSF-c) neuron with confocal microscopy in non-expanded spinal cord.

**Source data 2.** Quantification of a cilium diameter of cerebrospinal fluid-contacting (CSF-c) neuron with stimulated emission depletion (STED) microscopy in non-expanded spinal cord.

**Source data 3.** Quantification of a cilium diameter of cerebrospinal fluid-contacting (CSF-c) neuron with confocal microscopy in expanded spinal cord (expansion microscopy, ExM).

*Figure 5 continued on next page*

*Figure 5 continued*

**Source data 4.** Quantification of a cilia diameter of cerebrospinal fluid-contacting (CSF-c) neuron with stimulated emission depletion (STED) microscopy in expanded spinal cord (ExSTED).

**Source data 5.** Quantification of a primary cilium diameter of cerebrospinal fluid-contacting (CSF-c) neuron with 9 + 0 symmetry with stimulated emission depletion (STED) microscopy in expanded spinal cord (ExSTED).

**Source data 6.** Quantification of a motile cilium diameter of cerebrospinal fluid-contacting (CSF-c) neuron with 9 + 2 symmetry with stimulated emission depletion (STED) microscopy in expanded spinal cord (ExSTED).

**Figure supplement 1.** Three-dimensional (3D) expansion microscopy (ExM) of expanded cilia protruding to the central canal.

**Figure supplement 2.** Cilia diameter with different techniques: Box plot representation of cilia diameters, measured in the different conditions, confocal, stimulated emission depletion (STED), expansion microscopy (ExM), and ExSTED.

**Figure supplement 2—source data 1.** Cilia diameters measured in cerebrospinal fluid-contacting (CSF-c) neurons with the different techniques: confocal, stimulated emission depletion (STED), expansion microscopy (ExM), and ExSTED microscopy.

**Figure 5—video 1.** Three-dimensional (3D) ExSTED visualizes cerebrospinal fluid-contacting (CSF-c) neurons and their cilia within the 3D geometry of the central canal.

https://elifesciences.org/articles/73114/figures#fig5video1

cilia of dopaminergic CSF-c neurons were all motile cilia (9 + 2 symmetry) (*Figure 6L–S, T*). We found that somatostatin-expressing CSF-c neurons are sensory neurons expressing mainly sensory cilia. However, the motile cilia in both somatostatin-expressing and dopaminergic CSF-c neurons might be involved in contributing to the flow of CSF through the central canal (*Figure 6U*).

The sensory cilia are a potential location for pH-sensitive receptors (*Atkinson et al., 2019*; *Vina et al., 2015*). We have recently shown that in somatostatin-expressing CSF-c neurons, acidic and alkaline responses are mediated by the ASIC3 and possibly PKD2L1 channel, respectively (*Jalalvand et al., 2016a*; *Jalalvand et al., 2016b*). Interestingly, we could detect the expression of both ASIC3 and PKD2L1 on the cilia of spinal CSF-c neurons in mice (*Figure 6—figure supplement 1A–Q*). Besides, we visualized Arl13b, a ciliary protein on cilia that has high expression in sensory cilia (*Figure 6—figure supplement 1R–V*).

## CSF neurons have one and rarely two cilia

Another advantage of expansion microscopy (ExM) is that more specific morphological details of cells can be revealed, which cannot be detected in non-expanded samples. Using ExM, paired cilia of CSF-c neurons were visualized in some cases (*Figure 7*). As reported, somatostatin-expressing and dopaminergic CSF-c neurons mainly have one cilium, but we could in rare cases (*N* = 8) detect two cilia per bulb of these neurons by ExM (*Figure 7A–F*). To confirm the finding, we performed z-stack scanning to see the whole bulb and its cilia in 3D (*Figure 7—video 1*).

## Discussion

In this study, we uncovered the dense spatial organization of somatostatin- and dopamine-expressing CSF-c neurons along the central canal. ExLSM allowed us to visualize and quantify CSF-c neurons in a large volume of the spinal cord with a fourfold increase in resolution, necessary to distinguish and count each individual cell. There were approximately twice as many somatostatin CSF-c neurons as dopaminergic CSF-c neurons along the whole length of the spinal cord. One area, at the level of the rostral part of the dorsal fin had, however, a somewhat larger number of somatostatin CSF-c neurons than elsewhere. This is the same spinal cord area in which a dramatic decrease occurs of the serotonin/dopamine neurons that are located just ventral to the central canal (*Figure 8A*, *Schotland et al., 1995*; *Zhang et al., 1996*). Whether this is a coincidence or a related phenomenon remains to be elucidated.

Somatostatin and dopamine DCVs were quantified with STED microscopy (*Figure 8B*). The size distribution is in line with previous electron microscopy studies (*Schotland et al., 1996*). The STED data reveal the presence of DCVs in several CSF-c cellular subcompartments, including the bulb protrusion into the central canal, soma, and axonal branches. We could show a marked reduction of somatostatin DCVs number density in the soma of CSF-c neurons when exposed to acidic or alkaline pH, while there was no effect on the dopamine DCVs number density. This lack of change in the number density of DCVs in dopaminergic CSF-c neurons agrees with the present electrophysiological results showing that these cells are not activated by changes of the extracellular pH.

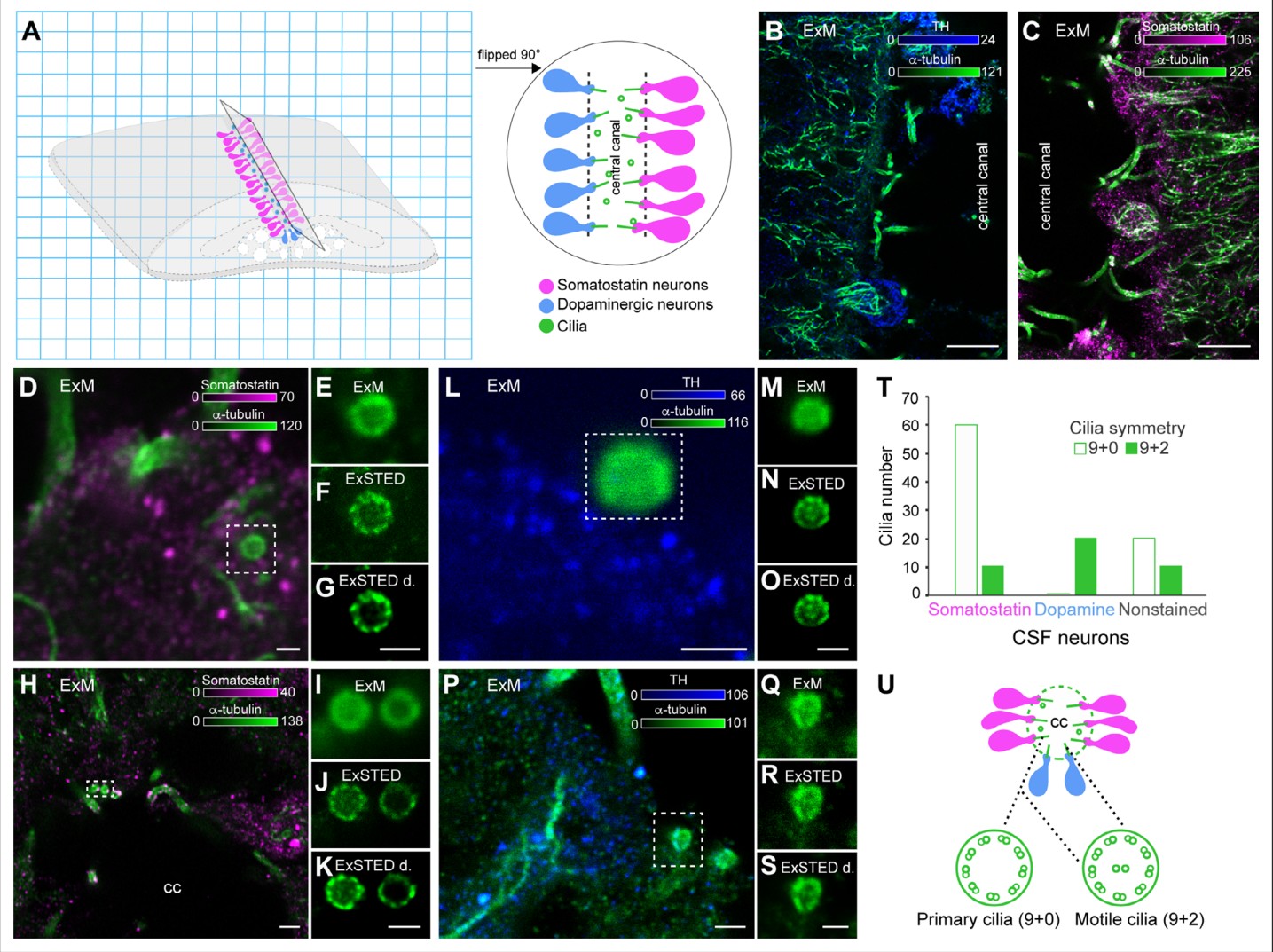

**Figure 6.** Cilia symmetries in somatostatin and dopaminergic cerebrospinal fluid-contacting (CSF-c) neurons. (**A**) A schematic illustration of an expanded spinal cord stained for somatostatin (magenta), dopamine (blue), and α-tubulin (green). The gel was cut through the central canal and flipped 90° on the side on the coverslip. (**B, C**) Longitudinal images of the expanded spinal cord (expansion microscopy, ExM) stained with α-tubulin, TH, and somatostatin antibodies, respectively. Scale bar, 10 μm. (**D–K**) ExM and ExSTED images of two somatostatin CSF-c neurons with their cilia. Scale bar, D, 1 μm and H, 3 μm. (**F, G**) showing 9 + 0 symmetry and (**J, K**) showing both 9 + 0 and 9 + 2 symmetries. Scale bar, 1 μm. (**L–S**) ExM and ExSTED images of two dopaminergic CSF-c neurons (TH staining) with 9 + 2 symmetry. Scale bar, L–O, Q–S, 1 μm and P, 2 μm. (**T**) Quantification of cilium types in somatostatin and dopaminergic CSF-c neurons. (**U**) A schematic illustration of the central canal with somatostatin and dopaminergic CSF-c neurons and their possible cilia symmetries. cc, central canal.

The online version of this article includes the following figure supplement(s) for figure 6:

**Figure supplement 1.** Polycystic kidney disease 2-like 1 (PKD2L1), acid-sensing ion channel 3 (ASIC3), and ARL13b expression on cerebrospinal fluid-contacting (CSF-c) neurons on mouse spinal cord.

We have shown previously that somatostatin/GABA-expressing CSF-c neurons are responding to acidic and alkaline extracellular solutions (*Jalalvand et al., 2016a*; *Jalalvand et al., 2016b*) being part of a homeostatic system for reducing motor activity when the spinal cord/brain is exposed to changes of pH as due to metabolic or respiratory stress or intense muscle activity. We now show with STED imaging that the pH response resulted from somatostatin DCVs release, whereas GABA was unaffected.

At the ultrastructural level, GABAergic CSF-c neurons have been shown to have ciliated endings in the lamprey and other species (*Orts-Del'Immagine et al., 2014*; *Schotland et al., 1996*; *Vigh et al., 2004*). Since somatostatin/GABA and dopaminergic/GABA CSF-c neurons have similar responses to mechanical stimuli, while only the former responds to pH changes, the question arose if these two

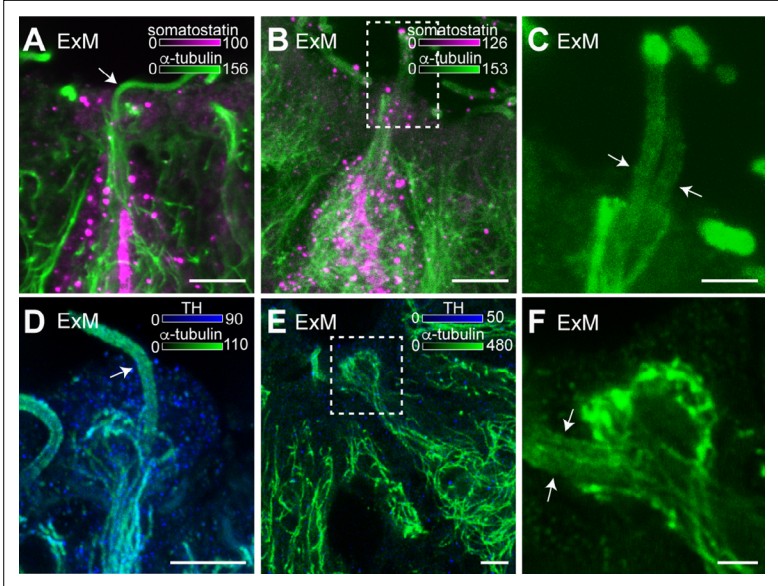

**Figure 7.** Cerebrospinal fluid-contacting (CSF-c) neurons have one or two cilia on their bulb protrusions. (**A**) An expansion microscopy (ExM) image of an expanded somatostatin CSF-c neuron showing one cilium on its bulb (arrow). Scale bar, 5 μm. (**B**, **C**) A somatostatin CSF-c neuron with two cilia (arrow) (**C**) selected ROI from (**B**). Scale bar B, 5 μm and C, 2 μm. (**D**) A dopaminergic CSF-c neuron (tyrosine hydroxylase [TH] expressing) with one cilium (arrow). Scale bar, 5 μm. (**E**, **F**) A dopaminergic CSF-c neuron with two cilia (arrows) (**F**) selected ROI from (**E**). Scale bar, E, 5 μm and F, 2 μm. cc, central canal.

The online version of this article includes the following video for figure 7:

**Figure 7—video 1.** Cerebrospinal fluid-contacting (CSF-c) neurons might contain two cilia.

https://elifesciences.org/articles/73114/figures#fig7video1

types of CSF-c neurons have different forms of cilia. To visualize cilia microtubule symmetry in CSF-c neurons a resolution of 5–10 nm is needed, which was provided by the combination of expansion and STED microscopy (ExSTED) (*Gao et al., 2018*). ExSTED microscopy allowed us to obtain cell-type-specific structural insights on the cilium types in ciliated somatostatin- and dopamine-expressing CSF-c neurons, respectively (*Figure 8B*). Using ExSTED microscopy we found primary cilia with 9 + 0 symmetry in somatostatin-expressing CSF-c neurons, also found in specialized sensory cells (*Kirschen and Xiong, 2017*; *Singla and Reiter, 2006*) and motile cilia with 9 + 2 symmetry in dopaminergic CSF-c neurons that may serve to generate fluid flow (*Faubel et al., 2016*; *Satir and Christensen, 2007*). The high expression of sensory cilia (9 + 0) found in pH/mechanosensitive somatostatin-expressing CSF-c neurons provide additional evidence to recognize them as sensory neurons. The mechanosensitivity of the dopamine- and somatostatin-expressing CSF-c neurons is mediated by different cellular mechanisms, the latter is blocked by APETx2, a selective ASIC3 blocker as shown in previous studies (*Jalalvand et al., 2016a*; *Jalalvand et al., 2016b*). The mechanosensitivity of the dopamine CSF-c neurons is, however, unaffected by APETx2. This blocker is specific to the ASIC3 but does not block ASIC1 receptors (*Diochot et al., 2004*), the latter having been cloned in lamprey (*Coric et al., 2005*) and found insensitive to pH. The data support the previous interpretation that ASIC3 mediates both the acid-sensing and the mechanosensitivity, in somatostatin-expressing CSF-c neurons (*Jalalvand et al., 2016a*; *Jalalvand et al., 2016b*). PKD2L1 channel is expressed in the dopaminergic CSF-c neurons (*Figure 4—figure supplement 2*) and this channel is known to serve as a mechanosensitive receptor (*Kamura et al., 2011*; *Nauli et al., 2003*) and may therefore serve in this function also for the dopaminergic CSF-c neurons. In the zebrafish, the mechanosensitive GABA CSF-c neurons sense the lateral bending movements occurring during locomotion through PKD2L1 channels (*Fidelin et al., 2015*). We have also observed expression of ASIC3 and PKD2L1 channels, as well as sensory cilia in CSF-c neurons of the mouse, which suggests that this may be representative of all vertebrates. The motile cilia (9 + 2) found only in the mechanosensitive (non-pH) dopaminergic CSF-c neurons may suggest that they may act as a CSF flow generator.

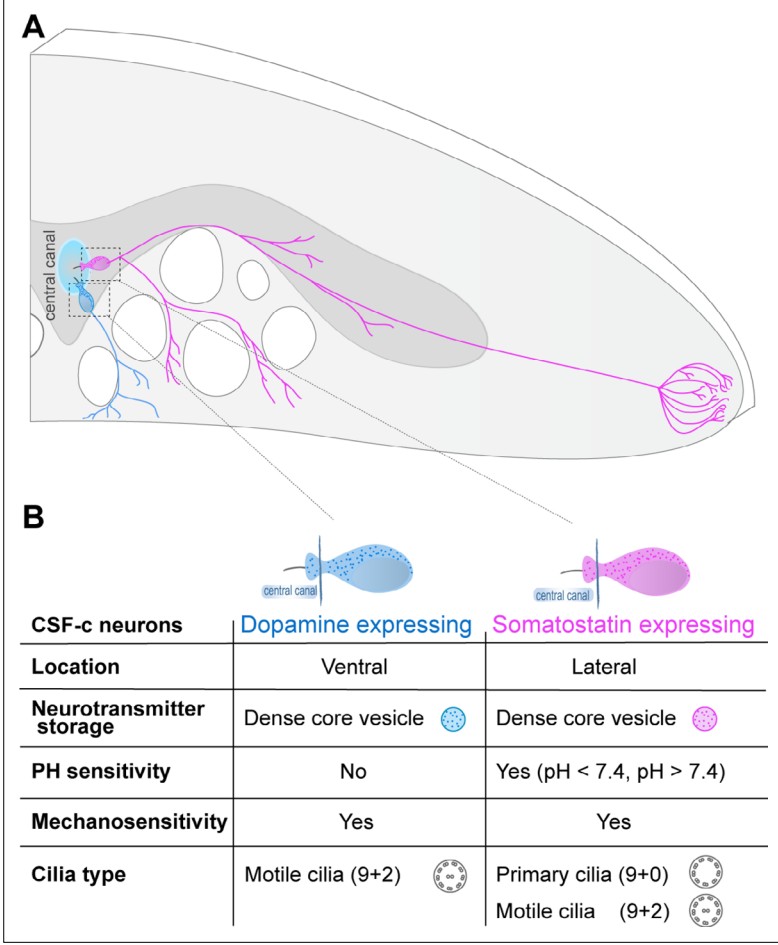

**Figure 8.** Somatostatin and dopaminergic cerebrospinal fluid-contacting (CSF-c) neurons are two distinct cell types with contrasting function along the spinal cord. (**A**) Schematic illustration of a cross-section of the lamprey spinal cord, with a somatostatin- and dopamine-expressing CSF-c neuron at the central canal and their axonal projections. (**B**) Summary of phenotypes, organelles, and physiological properties of dopaminergic and somatostatin CSF-c neurons.

In conclusion, by applying ExM combined with light-sheet and STED microscopy with nanoscale precision, the spatial organization, abundance, and subcellular composition of two distinct GABAergic CSF-c neuronal subtypes in the lamprey spinal cord were elucidated. One type is sensitive to pH and displaying mechanosensitivity (somatostatin) while the other (dopamine) only responds to mechanical stimuli. They use different cellular mechanisms for the transduction of mechanical stimuli to their cilia.

# Materials and methods

**Key resources table**

| Reagent type (species) or resource | Designation | Source or reference | Identifiers | Additional information |
|---|---|---|---|---|
| Biological sample (*Lampetra fluviatilis*) | Spinal cord | Collected from the Ljusnan River, Hälsingland, Sweden | | Freshly isolated from *Lampetra fluviatilis* |
| Biological sample (*Mus musculus*) | Spinal cord | Janvier Labs, C57BL/6 | | Freshly isolated from C57BL/6 |

*Continued on next page*

*Continued*

| Reagent type (species) or resource | Designation | Source or reference | Identifiers | Additional information |
|---|---|---|---|---|
| Antibody | Anti-acetylated tubulin (mouse monoclonal) | Sigma-Aldrich | Cat# T6793, RRID:AB_477585 | IF (1:500) |
| Antibody | Anti-somatostatin (rat monoclonal) | Millipore | MAB354, RRID:AB_2255365 | IF (1:100) |
| Antibody | Anti-somatostatin 14-IgG (rabbit polyclonal) | Peninsula laboratories | Cat# T-4102.0400, RRID:AB_518613 | IF (1:1000) |
| Antibody | Anti-TH (mouse monoclonal) | Millipore | Cat# MAB318, RRID:AB_2201528 | IF (1:200) |
| Antibody | Anti-TH (rabbit polyclonal) | Millipore | Cat# AB152, RRID:AB_390204 | IF (1:500) |
| Antibody | Anti-dopamine (mouse monoclonal) | Millipore | Cat# MAB5300, RRID:AB_94817 | IF (1:400) |
| Antibody | Anti-GABA (mouse monoclonal) | Swant | Cat# Mab 3A12, RRID:AB_2314454 | IF (1:2000) |
| Antibody | Anti-polycystin-L (rabbit polyclonal) | Millipore | Cat# AB9084, RRID:AB_571091 | IF (1:500) |
| Antibody | Anti-ASIC3 (rabbit polyclonal) | Thermo Fisher Scientific | Cat# PA5-41022, RRID:AB_2576713 | IF (1:400) |
| Antibody | Anti-ARL13B (rabbit polyclonal) | Proteintech | Cat# 17711-1-AP, RRID:AB_2060867 | IF (1:500) |
| Antibody | Donkey-anti-rat-IgG-AF594 | Jackson ImmunoResearch Labs | Cat# 712-585-153, RRID:AB_2340689 | IF (1:500) |
| Antibody | Donkey-anti-rat-IgG-AF488 | Jackson ImmunoResearch Labs | Cat# 712-545-153, RRID:AB_2340684 | IF (1:200) |
| Antibody | Goat-anti-mouse-STAR635P | Abberrior | Cat# ST635P-1001-500 UG, RRID:AB_2893232 | IF (1:500) |
| Antibody | Goat-anti-rabbit-AF594 | Thermo Fisher Scientific | Cat# A-11037, RRID:AB_2534095 | IF (1:500) |
| Antibody | Donkey-anti-mouse-IgG-Cy3 | Jackson ImmunoResearch Labs | Cat# 715-165-150, RRID:AB_2340813 | IF (1:500) |
| Antibody | Donkey-anti-mouse-IgG-AF488 | Jackson ImmunoResearch Labs | Cat# 715-545-150, RRID:AB_2340846 | IF (1:200) |
| Other | NeuroTrace530/615 | Thermo Fisher Scientific | Cat# N21482, RRID:AB_2620170 | IF (1:1000) |
| Other | NeuroTrace640/660 | Thermo Fisher Scientific | Cat# N21483, RRID:AB_2572212 | IF (1:1000) |
| Other | Phalloidin-STAR635P | Abberior | | IF (1:200) |
| Peptide, recombinant protein | Neurobiotin | Vector Laboratories | Cat# SP-1120, RRID:AB_2313575 | Injection of 0.5% solution for intracellular labelling |
| Peptide, recombinant protein | Streptavidin-AF488 | Jackson ImmunoResearch | Cat# 016-540-084, RRID:AB_2337249 | IF (1:1000) |
| Chemical compound, drug | Glutamate receptor antagonist kynurenic acid | Tocris Ellisville, MO, USA | | Bath perfusion, 2 mM |
| Chemical compound, drug | GABAA receptor antagonist gabazine | Tocris Ellisville, MO, USA | | Bath perfusion, 20 mM |
| Software, algorithm | Fiji | *Schindelin et al., 2012* | RRID:SCR_002285 | |

*Continued on next page*

*Continued*

| Reagent type (species) or resource | Designation | Source or reference | Identifiers | Additional information |
|---|---|---|---|---|
| Software, algorithm | MATLAB | The Mathworks | RRID:SCR_001622 | |
| Software, algorithm | Impsector | Max-Planck Innovation | RRID:SCR_015249 | |
| Software, algorithm | Origin | OriginLab | RRID:SCR_014212 | |
| Software, algorithm | Imaris 9.1 | Bitplane | RRID:SCR_007370 | |
| Software, algorithm | Clampex and Clampfit | Molecular Devices, CA, USA | RRID:SCR_011323 | |
| Commercial assay or kit | Digoxigenin RNA Labeling kit | Roche Diagnostics | Catalog #11 277 073 910 | In situ hybridization |
| Commercial assay or kit | TSA Cy3 Plus Evaluation Kit | PerkinElmer | NEL763E001 | In situ hybridization |
| Antibody | Anti-DIG antibody coupled to HRP (sheep polyclonal) | Roche Diagnostics | RRID: AB_514497 | IF (1:2000) |

## Animals: lamprey

Experiments were performed on a total of 40 adult river lampreys (*Lampetra fluviatilis*) of both sexes that were collected from the Ljusnan River, Hälsingland, Sweden. The experimental procedures were approved by the local ethical committee (Stockholms Djurförsöksetiska Nämnd; Dnr 5806-2019) and were in accordance with The Guide for the Care and Use of Laboratory Animals (National Institutes of Health, 1996 revision). During the investigation, every effort was made to minimize animal suffering and to reduce the number of animals used during the study.

## Mouse

Experiments were performed on a total of 4, C57BL/6 wild type mice (*Mus musculus*). All experiments were performed in accordance with animal welfare guidelines set forth by Karolinska Institutet and were approved by Swedish Board of agriculture for Animal welfare (ethical permit number: 2645-2021).

## Immunohistochemistry: lamprey

The animals ($n$ = 30) were deeply anesthetized through immersion in carbonate-buffered tap water containing MS-222 (100 mg/l; Sigma, St Louis, MO, USA). Following decapitation, a portion of the spinal cord, rostral to the dorsal fin, was fixed by 4% (wt/vol) paraformaldehyde (PFA) in phosphate-buffered saline (PBS) for 12–24 hr at 4°C, and subsequently cryoprotected in 20% sucrose in phosphate buffer (PB) for 3–12 hr. For GABA and dopamine immunodetection, 1% glutaraldehyde (vol/vol) was added to the fixative solution. Transverse sections (20 µm thick) were cut on a cryostat (Microm HM 560, Microm International GmbH, Walldorf, Germany), collected on gelatine-coated slides, and kept at −20°C until processing. Sections were incubated/co-incubated with different primary antibodies listed here: a mouse monoclonal anti-acetylated tubulin antibody (dilution 1:500, Sigma-Aldrich), a rat monoclonal anti-somatostatin antibody (1:100, Millipore), a rabbit polyclonal anti-somatostatin-14 IgG antibody (1:1000, Peninsula laboratories), a mouse monoclonal anti-TH antibody (1:200, Millipore), a rabbit polyclonal anti-TH antibody (1:500, Millipore), a mouse monoclonal anti-dopamine antibody (1:400, Millipore), and a mouse monoclonal anti-GABA antibody (1:2000, Swant).

## Mouse

The animals were deeply anesthetized with sodium pentobarbital (200 mg/kg i.p.) and transcardially perfused with 4% PFA in 0.01 M PBS, pH 7.4. The spinal cord was removed and postfixed for 2 hr, after which it was transferred to a 12% sucrose solution in 0.01 M PBS overnight. Transverse sections (20 µm thick) were cut on a cryostat and mounted on gelatine-coated slides and kept at −20°C until processing. Sections were incubated with a rabbit polyclonal anti-polycystin-L antibody (1:500, Millipore), a rabbit polyclonal ASIC3 antibody (1:400, Thermo Fisher Scientific), or a rabbit polyclonal anti-ARL13B antibody (1:500, Proteintech).

The lamprey and mouse spinal cord sections were after incubation with primary antibodies thoroughly rinsed in PBS and then incubated with Alexa Fluor 594-conjugated donkey anti-rat IgG (1:500, Jackson ImmunoResearch Labs) or Alexa Fluor 488-conjugated donkey anti-rat IgG (1:200, Jackson

ImmunoResearch Labs), STAR 635P-conjugated goat anti-mouse (1:500, Abberior) or Alexa Fluor 594-conjugated donkey anti-mouse (1:500, Thermo Fisher Scientific), and STAR 635P-conjugated goat anti-rabbit (1:500, Abberior) or Alexa Fluor 594-conjugated goat anti-rabbit (1:500, Thermo Fisher Scientific), for 3 hr at room temperature. The sections were Nissl stained by adding NeuroTrace 530/615 red or 640/660 deep-red fluorescent Nissl stain (1:1000, Invitrogen) to the secondary antibody solution. Phalloidin conjugated with STAR 635P (1:200, Abberior) was added to the secondary antibodies to stain actin filaments. The primary and secondary antibodies were diluted in 1% bovine serum albumin and 0.3% Triton X-100 in 0.1 M PB. All sections were mounted in custom-made Mowiol mounting media, supplemented with DABCO (Thomas Scientific, C966M75), and covered with coverslips (No. 1.5).

## Expansion microscopy

After fixation, transverse sections of the lamprey spinal cord (40–50 µm thick) were cut on a cryostat and collected and immersed in PBS. The sections were stained with antibodies according to the protocols described above. The samples were treated at room temperature for 1 hr with anchoring solution, 1 mM MA-NHS in PBS (0.018 g MA-NHS in 100 µl DMSO and stored at −20°C) which enables proteins to be anchored to the hydrogel. The slices then were incubated for 1 hr in a monomer solution on ice, which was followed by adding the gelling solution to the gelling chambers. The gelling solution consisted of monomer solution (1 M NaCl, 8.6% sodium acrylate, 2.5% acrylamide, and 0.15% $N,N'$-methylene bisacrylamide in PBS), 4-hydroxy-TEMPO (0.2%), Tetramethylethylenediamine (TEMED) (accelerator solution, 0.2%), and Ammonium persulfate (APS) (initiator solution, 0.2%). The slices were incubated in a 37°C incubator for 2 hr for gelation. The gels were removed from the gelling chamber and the coverslips were transferred to digesting buffer (50 mM Tris, pH 8.0, 1 mM EDTA, and 0.5 Triton X-100) with proteinase K (1:100, 8 units/ml, New England Biolabs) for 30–40 min in room temperature. The gels were removed from the digestion buffer and immersed in deionized (DI) water (three to five times for 30 min) for further expansion. After final expansion, the gels were mounted to a 35 mm glass bottom petri dish coated with poly-L-lysine (Sigma-Aldrich). To remove the extra gel on a coverslip and increase the resolution, we cut the gel through the central canal and rotated it 90° away from the coverslip. The expansion factor was ~4.5–5 and has been calculated by overlapping the pre- and post-expanded gel slice in the air–water boundary and by cross checking the cilia diameter in expanded and not samples.

## STED microscopy

Most STED images have been recorded on a custom-built setup, as previously described (*Alvelid and Testa, 2019*) equipped with a glycerol objective (HCX PL APO 93×/1.30 NA GLYC STED White motCORR, 15506417, Leica Microsystems). The images were recorded by exciting Alexa Fluor 594 and STAR 635P with 561 nm (PDL561, Abberior Instruments) and 640 nm (LDH-D-C-640, PicoQuant) laser lines, respectively. A STED beam at 775 nm (KATANA 08 HP, OneFive) has been used to deplete both laser lines, shaped by a spatial light modulator (LCOS-SLM X10468-02, Hamamatsu Photonics) into a donut. Two-colour STED images were recorded line-by-line sequentially. A third confocal channel, for structures labelled with Alexa Fluor 488, has been excited with a 510-nm laser line (LDH-D-C-510, PicoQuant). Detection was performed using the following bandpass filters: ET615/30 m (Chroma), ET705/100 m (Chroma), and FF01-550/49 (Semrock). The pixel sizes of the images were 27.2–32.3 nm in XY, and 300 nm in Z. The pixel dwell time used was 10–50 µs, added over one to two lines. Laser powers used were in the following ranges: 561 nm: 1–40 µW, 640 nm: 2–20 µW, and 775 nm: 65–180 mW. Additional STED images (*Figures 2 and 3A–J*) have been recorded on a commercial Leica TCS SP8 3X STED. The images were recorded by exciting Alexa Fluor 488, Alexa Fluor 594, and STAR 635P with laser lines at 488, 594, and 635 nm, respectively. The detection windows used were 510–560, 600–645, and 670–750 nm. The pixel size of the images was 25–30 nm, and the pixel dwell time used was 10–40 µs, added over two to four lines.

## Light-sheet microscopy

The expanded slices from four different parts of the lamprey spinal cord were separately glued to a metal rod and placed in the chamber of a Zeiss Light-sheet Z1 microscope containing DI water.

Fluorescence was excited from two sides using ×10/0.2 NA illumination objectives and detected using a ×10/0.4 NA or ×20/1.0 NA water dipping objective.

## Image analysis

The images were analysed using Fiji (*Schindelin et al., 2012*), Imspector (Max-Planck Innovation), and MATLAB (The Mathworks). The transverse sections of the cilia images were deconvoluted with a calculated effective STED PSF, Lorentzian with FWHM of 50 nm, using the Richardson–Lucy deconvolution implemented in Imspector. The deconvolution was stopped after five iterations. The line profiles of cilia and DCVs were fitted with a Gaussian, and the FWHM was measured. The OriginLab software (OriginLab) was used for making the graphs. Animation and spot object creation tools of Imaris 9.1 (Bitplane) were used to make 3D movies and segmentation of the 3D data.

Analysis of the correlation of GABA signal and somatostatin DCVs (*Figure 3v*) was performed by manually selecting areas inside (soma/axon-DCVs) and outside (soma/axon-volume) somatostatin DCVs (detected in the somatostatin channel) as well as in the extracellular space (Background). The mean of the GABA signal inside each area was recorded. A total of five cells were used for the soma and three cells for the axons. The number of areas selected in each category in each cell ranged from 3 to 63, with a mean of 37. Plotted are the paired graphs for each cell, with each data point representing the cellular mean of the mean GABA signals per area. The Student's *t*-test is performed between the groups of means of the cells per category.

The quantification of the somatostatin and dopamine DCVs in the cell area (2D) was performed using Fiji Analyze particles plugin. Vesicles with an area smaller than 0.190 $\mu m^2$ (corresponding to vesicle diameter less than 250 nm) were considered and their number density was calculated.

The quantification of the somatostatin and dopamine DCVs in cell volume was performed using the 3D imageJ suite plugins (*Ollion et al., 2013*). The 3D stacks were slightly adjusted for noise applying a two-pixel 3D median filter. The background was detected by applying a 15-pixel 3D median filter and then subtracted from the stacks. The stacks were then segmented by first detecting the seeds points using the maximum local filter and then applying the 3D spot segmentation. Vesicles with a volume smaller than 0.008 $\mu m^3$ were considered for the final quantification and their number density was calculated.

## Live slices

The spinal cord of lamprey (*n* = 5) was dissected out and embedded in 4% agar dissolved in ice-cooled oxygenated 4-(2-hydroxyethyl)-1-piperazineethanesulfonic acid (HEPES)-buffered physiological solution containing (in mM): 138 NaCl, 2.1 KCl, 1.8 $CaCl_2$, 1.2 $MgCl_2$, 4 glucose, 2 HEPES, and with pH adjusted to 7.4 with NaOH. The agar block containing the spinal cord was glued to a metal plate and transverse slices of the spinal cord (100–150 or 300 μm) were cut on a vibrating microtome. The preparation was continuously perfused with HEPES solution at 4–6°C. Then the spinal cord slice was exposed to HEPES solution with various pH values (7.4, 6.5, and 8.5) for 8–10 min. The slices were then fixed immediately with 4% PFA (somatostatin) or 4% PFA and 1% glutaraldehyde (dopamine and GABA) in 0.01 M PBS overnight at 4°C. Following a thorough rinse in 0.01 M PBS, the slices (100–150 μm) were incubated with a rat monoclonal anti-somatostatin, a mouse monoclonal anti-GABA, or a mouse monoclonal anti-dopamine antibody overnight at 4°C. The slices were then incubated with Alexa Fluor 594-conjugated donkey anti-rat or anti-mouse IgG as described above.

## Patch-clamp recordings

The spinal cord slices (300 μm) of lamprey (*n* = 5) after slicing were transferred in a cooled recording chamber and allowed to recover at 5°C for 1 hr before recording. Patch electrode was prepared from borosilicate glass microcapillaries (Hilgenberg GmbH) using a two-stage puller (PP830, Narishige, Japan). Patch electrodes (8–12 MΩ) were filled with an intracellular solution of the following composition (in mM): 130 K-gluconate, 5 KCl, 10 HEPES, 4 Mg-ATP, 0.3 Na-GTP, and 10 phosphocreatine sodium salt. The pH of the solution was adjusted to 7.4 with KOH and osmolarity to 270 mOs $ml^{-1}$ with water. Cells ventral to the central canal were recorded in whole-cell in current-clamp mode using a Multiclamp 700B amplifier (Molecular Devices Corp., CA, USA). Bridge balance and pipette capacitance compensation were adjusted and signals were digitized and recorded using Clampex software and analysed in Clampfit (pCLAMP 10, Molecular Devices, CA, USA). The neurons were visualized with

DIC/infrared optics (Olympus BX51WI, Tokyo, Japan). Resting membrane potentials were determined in current-clamp mode during whole-cell recording. The following drugs were added to the extracellular solution and applied by bath perfusion: the GABAA receptor antagonist gabazine (20 mM, Tocris, Ellisville, MO, USA) and the glutamate receptor antagonist kynurenic acid (2 mM, Tocris, Ellisville, MO, USA). Neurons were intracellularly labelled by injection of 0.5% Neurobiotin (Vector Laboratories) during whole-cell recordings. After recording the spinal cord slices were fixed with 4% formalin. To investigate whether the intracellularly Neurobiotin-labelled CSF-c cells express TH, the slices were incubated overnight with a mouse monoclonal anti-TH antibody and then rinsed thoroughly in 0.01 M PBS and incubated with a mixture of Alexa Fluor 488-conjugated streptavidin (1:1000, Jackson ImmunoResearch Labs) and Cy3-conjugated donkey anti-mouse IgG (1:500, Jackson ImmunoResearch Labs) for 3 hr at room temperature.

## In situ hybridization

Lamprey ($n = 2$) were deeply anesthetized as described above and the rostral spinal cord was removed, fixed in 4% formalin in 0.1 M PB overnight at 4°C, and then cryoprotected in 20% sucrose in 0.1 M PB. Then, 10–20 µm thick cryostat sections were made and stored at −80°C until processed. Single-stranded digoxigenin-labelled sense and antisense PKD2L1 riboprobes were generated by in vitro transcription of the previously cloned PKD2L1 cDNA using the Digoxigenin RNA Labeling kit (Roche Diagnostics). Briefly, sections were incubated for 1 hr in prehybridization mix (50% formamide, 5× SSC, 1% Denhardt's, 50 g/ml, salmon sperm DNA, 250 g/ml yeast RNA) at 60°C. Sections incubated with the heat-denatured digoxigenin-labelled riboprobe were hybridized overnight at 60°C. Following the hybridization, the sections were rinsed twice in 1× SSC, washed twice in 1× SSC (30 min each) at 60°C and twice in 0.2× SSC at room temperature. After blocking in 0.5% blocking reagent (PerkinElmer), the sections were incubated overnight in anti-DIG antibody coupled to HRP (1:2000, Roche Diagnostics) at 4°C. The probe was then visualized by TSA Cy3 Plus Evaluation Kit (PerkinElmer). The specificity of the hybridization procedure was verified by incubating sections with the sense riboprobe (data not shown). The sections were rinsed thoroughly in 0.01 M PBS and then incubated with a mouse monoclonal anti-TH antibody (1:200, Millipore) overnight at 4°C, rinsed in PBS, and incubated with Alexa Fluor 488-conjugated donkey anti-mouse IgG (1:200, Jackson ImmunoResearch Labs) and 640/660 deep-red fluorescent Nissl stain (1:1000, Thermo Fisher Scientific) for 2 hr and mounted with Mowiol. All primary and secondary antibodies were diluted in 1% BSA and 0.3% Triton X-100 in 0.1 M PB.

## Acknowledgements

We thank the ALM facility at the Science for Life Laboratory for access to the STED Leica Sp8. We thank the SciLifeLab SFO funding, the GGS foundation, and the Swedish Foundation for Strategic Research for supporting the project FFL15-0031, and grants from The Swedish Research Council to SG (2021-01995).

## Additional information

### Competing interests

Ilaria Testa: Reviewing Editor, eLife. The other authors declare that no competing interests exist.

### Funding

| Funder | Grant reference number | Author |
| --- | --- | --- |
| Swedish Foundation for Strategic Research | FFL15-0031 | Ilaria Testa |
| Swedish Research Council | 2021-01995 | Sten Grillner |
| Royal Institute of Technology | Göran Gustafssons Stiftelse | Ilaria Testa Elham Jalalvand |

| Funder | Grant reference number | Author |
|--------|------------------------|--------|

The funders had no role in study design, data collection, and interpretation, or the decision to submit the work for publication.

## Author contributions

Elham Jalalvand, Data curation, Formal analysis, Investigation, Methodology, Validation, Visualization, Writing - original draft, Writing - review and editing; Jonatan Alvelid, Giovanna Coceano, Steven Edwards, Formal analysis, Methodology, Writing - review and editing; Brita Robertson, Formal analysis, Writing - review and editing; Sten Grillner, Writing - review and editing; Ilaria Testa, Methodology, Supervision, Validation, Visualization, Writing - original draft, Writing - review and editing

## Author ORCIDs

Elham Jalalvand ⓘ http://orcid.org/0000-0003-4759-3301
Jonatan Alvelid ⓘ http://orcid.org/0000-0002-3554-9322
Sten Grillner ⓘ http://orcid.org/0000-0002-8951-3691
Ilaria Testa ⓘ http://orcid.org/0000-0003-4005-4997

## Ethics

All experiments were performed in accordance with animal welfare guidelines set forth by Karolinska Institutet and were approved by Stockholm North Ethical Evaluation Board for Animal Research.

## Decision letter and Author response

Decision letter https://doi.org/10.7554/eLife.73114.sa1
Author response https://doi.org/10.7554/eLife.73114.sa2

# Additional files

## Supplementary files

• Transparent reporting form

## Data availability

Data availability The data that support the implementation of the method and support the findings in this study, including images and statistical analysis are openly available in Zenodo with reference number 5758671 at https://doi.org/10.5281/zenodo.5758671.

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
