## [Editor Report]

Here, the authors use a variety of optical super-resolution techniques to explore the structure and function of different neurons in tissue. They present very interesting evidence that sensory neurons contacting the cerebrospinal fluid in lamprey differ in the motility of their cilium and in their response to variations of pH: while somatostatin-positive ciliated sensory neurons lose dense core vesicles from their soma and enrich them in their axons, dopaminergic ciliated sensory neurons do not show any change in DSV localization / density. This manuscript is of broad interest for the neuroscience and imaging community.

---

## [Decision Letter]

**Decision letter after peer review:**

Thank you for submitting your article "ExSTED microscopy reveals contrasting functions of dopamine and somatostatin CSF-c neurons along the central canal" for consideration by *eLife*. Your article has been reviewed by 3 peer reviewers, and the evaluation has been overseen by Suzanne Pfeffer as the Senior and Reviewing Editor. The following individuals involved in review of your submission have agreed to reveal their identity: Francesca Bottanelli (Reviewer #1); Claire Wyart (Reviewer #2).

Essential revisions:

1. The introduction should give more insights into the underlying biology and also clarify which questions are being asked here: what is new and why these are important, to help the reader understand the biology that underlies this work.

2. The interpretation of physiological results based on established knowledge of ASICs in lamprey cannot be easily explained, and the authors need to provide further data to support their claims or rephrase their conclusions with great care. Specific details follow in the individual reviews below. Electrophysiological recordings of both SST+ and DA+ neurons is important to make conclusions related to the functional roles of these cells, as the microscopy alone cannot infer on the sensory function; alternatively the text needs to be modified in line with suitable conclusions from the morphological analyses.

3. Many of the figures were provided in suboptimal resolution and it is essential to be sure they are provided at the appropriate quality.

*Reviewer #1 (Recommendations for the authors):*

The work is generally well presented. However, some additional information could make it easier for the non-expert reader. Here are some suggestions:

• A model at the end to compile all results on pH sensitivity, mechanosensing, etc… would greatly simplify the understanding of the messages of the paper.

• TH staining is not explained.

• The authors claim a resolution below 10 nm however the size of the antibodies used for staining should be considered.

• The authors say "Somatostatin-expressing CSF-c neurons are found throughout the whole volume and located laterally relative to the central canal, while the dopaminergic CSF-c neurons are located ventrally" could the author point that out in figure 1 as I have a hard time visualizing it.

• Figure 1f = Cell number per volume? Or total cell number?

• Figure 3g = vesicles per volume? Or total amount of vesicles per soma? Additionally, the increase of number of vesicles in axon is not quantified?

• Figure 4d = same comment as for figure 3.

• Figure 5 = I am wondering why the expansion factor was calculated by measuring the gel slice rather than looking at a sub-cellular structure. Could the authors calculate the expansion factor by measuring cilia diameter in STED and expanded confocal samples?

*Reviewer #2 (Recommendations for the authors):*

1. On the microscopy section, the authors make a statement that DSVs within somatostatin-positive sensory cells are present in the axons at baseline and become more numerous upon pH variations. To illustrate their point, the authors need to show:

a) Images illustrating that DSVs were found in the axonal extension (page 5, "data not shown") at baseline, and;

b) Evidence determining whether DSVs are released or transported from the soma to the axon (Figure 3d-f) upon changes of pH. A quantification of the total number of DSV per cell could be insightful to resolve these 2 interpretations.

This point is important for interpretation of the mechanisms at play upon variations of pH.

2. The interpretation of physiological results based on established knowledge of ASICs in lamprey cannot be easily explained, and the authors need to provide further data to support their claims or rephrase their conclusions:

a) the authors need to show that somatostatin-positive cells respond to pH in their conditions as it is key for their interpretation, and their previous article on all spinal CSF-contacting neurons did not mention that only half the ciliated neurons contacting the CSF responded to pH (Jalalvand et al. Current Biology 2016).

b) Similarly, the authors here examine the mechanosensory response of dopaminergic CSF-contacting neurons without quantifying the response of somatostatin-positive neurons to the same stimulus. Both responses to variations of pH and mechanical stimulation need to be performed in both cell types.

c) Another group have shown that lamprey only express one type of ASIC channel, ASIC1 (*not* the channel ASIC3 ): see https://www.ncbi.nlm.nih.gov/pmc/articles/PMC3047259/). Therefore, how can the authors explain the effect of the drug used against ASIC3 in this context ?

In order to solve this issue, the authors could show expression of ASIC channels, and test the effect of a drug against ASIC1 on the response of somatostatin-expressing cells.

d) Finally, the only ASIC channel expressed in lamprey, ASIC1, when cloned and tested, has been reported to be proton insensitive (see https://www.ncbi.nlm.nih.gov/pmc/articles/PMC1464184/)--how then to explain then the activation of a subset of ciliated neurons exposed to protons?

[Editors' note: further revisions were suggested prior to acceptance, as described below.]

Thank you for resubmitting your work entitled "ExSTED microscopy reveals contrasting functions of dopamine and somatostatin CSF-c neurons along the lamprey central canal" for further consideration by *eLife*. Your revised article has been reviewed by 3 peer reviewers and the evaluation has been overseen by Suzanne Pfeffer as the Senior and Reviewing Editor.

We would like to publish this story in *eLife* but ask you to please make textual changes in response to this reviewer's comments.

Specific comments from one reviewer:

The authors satisfactorily answered some of the issues raised on the quantification of images and provided high resolution images for their outstanding data acquired at high resolution. These data are very convincing and beautiful. Congratulations, this will be very precious and inspirational for further studies in the spinal field !

The issue remaining are related to the interpretation of the physiological results. The authors should either show expression of ASIC3 in CSF-cNs by ISH or tone down their conclusions relative to the ASIC3 receptor in response to pH.

Similarly the authors should be more careful for stating the role PKD2L1 in pH sensing or mechanoreception as they lack genetic methods to investigate its function in lamprey.

A) On the role of ASIC3 in pH sensing :

As stated in my first review, ASIC3 can only be found in the genome of mammals. Only ASIC1 has been identified in the lamprey genome (https://www.ncbi.nlm.nih.gov/pmc/articles/PMC3047259) and the lamprey ASIC1a is proton insensitive (https://www.ncbi.nlm.nih.gov/pmc/articles/PMC1464184/).

The authors did not respond to my request of testing the role of ASIC1a in their experiments using pharmacology.

One can therefore wonder whether 2uM APETx2 used by the authors (about 100X the EC50 of ASIC3 homomers, https://pubmed.ncbi.nlm.nih.gov/17113616/) could be acting on other targets (see suggested actions on Na or K channels mentioned here, https://pubmed.ncbi.nlm.nih.gov/17113616/). Please discussion clearly in the text.

B) On the role of PKD2L1 in mechanoreception:

The authors summarize that PKD2L1 is responsible for mechanoreception only in dopaminergic CSF-cNs.

However:

1. PKD2L1 is expressed in all CSF-cNs, ventral (dopaminergic) and dorsal (somatostatinergic) as shown by the authors here and found in zebrafish (Djenoune 2014) and mouse (Huang 2006; Petracca 2018). In fact, in their 2016b publication, the authors had proposed that SST+ CSF-cNs were responding to basic pH via PKD2L1, suggesting a role for this channel in these cells.

Why therefore in this study, only mentioning the role of PKD2L1 for mechanoreception in the dopaminergic CSF-cNs despite the expression being there in both dorsal and ventral cell types?

2. Due to the lack of specific antagonists, the authors do not have the tools in lamprey to measure its contribution to mechanoreception. They can only suggest that PKD2L1 contribute to mechanoreception in both ventral and dorsal CSF-cN types.

3. Please correct an error in citation and references used:

The authors use Bohm et al. 2016 to state that CSF-cNs rely on PKD2L1 to be mechanosensory in zebrafish.

To be correct, we showed in vivo in Bohm 2016 that both ventral and dorsal CSF-cNs respond to concave (not convex) mechanical deformations of the spinal cord via PKD2L1 (response are abolished in the KO).

However, mechanoreception cannot be rigorously demonstrated in vivo. It is therefore only in Sternberg 2018 that we could show in vitro using a piezo device to mechanically stimulate their membrane that all CSF-cNs isolated in primary cultures are mechanosensory cells and that their response always rely on PKD2L1.

Note that in zebrafish, we now understand that CSF-cNs in vitro do not respond to CSF flow (Prendergast et al. under review) and in vivo, their response to concave mechanical bending of the spinal cord needs their interaction with the Reissner fiber (Orts Dell Immagine 2020), which does not alter itself the flow (Cantaut-Belarif 2018).

The authors should at minima cite Sternberg et al. 2018 for showing the role of PKD2L1 in mechanoreception, but to be fair, also propose with more nuances in the discussion how these cells in lamprey can combine ASIC and PKD channels to sense pH and mechanical inputs, citing Bohm 2016, Sternberg 2018 and Orts Del Immagine 2020.

---

## [Author Response]

Essential revisions:1. The introduction should give more insights into the underlying biology and also clarify which questions are being asked here: what is new and why these are important, to help the reader understand the biology that underlies this work.

We have rewritten part of the Introduction to better expose the underlying biological questions.

2. The interpretation of physiological results based on established knowledge of ASICs in lamprey cannot be easily explained, and the authors need to provide further data to support their claims or rephrase their conclusions with great care. Specific details follow in the individual reviews below. Electrophysiological recordings of both SST+ and DA+ neurons is important to make conclusions related to the functional roles of these cells, as the microscopy alone cannot infer on the sensory function; alternatively the text needs to be modified in line with suitable conclusions from the morphological analyses.

We have previously reported on the sensitivity of the somatostatin-CSF-contacting neurons (SST+) specifically showing that they respond to both fluid movements and pH changes by electrophysiology (Jalalvand et al. 2016, Nature Communications, Jalalvand et al. 2016, current Biology). We now performed similar experiment on dopaminergic CSF-c neurons in the current study. Since the responses of SST+CSF neurons have been quantified in considerable detail earlier, we feel that there would be no reason to repeat these results in the context of these experiments.

3. Many of the figures were provided in suboptimal resolution and it is essential to be sure they are provided at the appropriate quality.

We provide Figures at high resolution.

Reviewer #1 (Recommendations for the authors):The work is generally well presented. However, some additional information could make it easier for the non-expert reader. Here are some suggestions:• A model at the end to compile all results on pH sensitivity, mechanosensing, etc… would greatly simplify the understanding of the messages of the paper.

A new figure, Figure 8, has been added to summarise our findings.

• TH staining is not explained.

Dopaminergic CSF-c neurons express Tyrosine hydroxylase (TH). TH catalyzes the hydroxylation of the Tyrosine to L-DOPA (precursor of Dopamine). Anti-TH antibody has been used to detect TH and is as a marker of dopaminergic neurons. In the Method/Immunohistochemistry section we have explained the type of used TH antibodies and the method of the staining.

• The authors claim a resolution below 10 nm however the size of the antibodies used for staining should be considered.

The 10 nm value was derived by the response of our STED system to point-like objects, such as single antibodies, which are imaged with a FWHM of ~35 10 nm, and further divided by the expansion factor of 4. However, this value refers to how good the response is, which is only one metric to define spatial resolution. When considering biological structures, we agree that the size of the labels indeed matters and influences the resulting ability of resolving and visualizing fine structures. In this work, to be able to tell apart the duplets, we needed at least a resolution of 10-15 nm, which was achieved with ExSTED but not with only STED.

• The authors say "Somatostatin-expressing CSF-c neurons are found throughout the whole volume and located laterally relative to the central canal, while the dopaminergic CSF-c neurons are located ventrally" could the author point that out in figure 1 as I have a hard time visualizing it.

We added in Figure 1 the annotation for dorsal and ventral to guide the viewer to the geometry of the central canal.

• Figure 1f = Cell number per volume? Or total cell number?

We have counted somatostatin and dopamine CSF-c cells in volume of 1.8 × 10^7^ µm^3^ in each part of the spinal cord. What we see in the graph shows the mean number of the respected cells per volume of each part. We have counted a total of 224 cells in 4 distinct parts of the spinal cord within a volume of 7.2 × 10^7^ µm^3^. (Information added in figure 1 legend)

• Figure 3g = vesicles per volume? Or total amount of vesicles per soma? Additionally, the increase of number of vesicles in axon is not quantified?

We have clarified the quantification presented in figure 3g in the figure legend. The somatostatin DCVs number density (µm^-2^) is reported per cell soma in N = 13 cells. These are 2DSTED images with a pixel size of 20-25 nm and an axial resolution of ~500 nm. (Information added in the Figure legend). We have additionally applied volumetric imaging with 2D STED and quantified somatostatin and dopamine DCVs densities as the number N within the soma volume (N x µm^-3^) (added in Figure 3—figure supplement 1 and Figure 4—figure supplement 1). Quantification of the density of vesicles in axons have not been included since we found it hard to have a comprehensive view of the entire axonal network.

• Figure 4d = same comment as for figure 3.

As above we quantified vesicle density in soma areas (N x µm^-2^) (2DSTED), and quantification vesicle number density of dopamine DCVs in volume (N x µm^-3^) has been added in Figure 4—figure supplement 1 (Information added in the Figure legend).

• Figure 5 = I am wondering why the expansion factor was calculated by measuring the gel slice rather than looking at a sub-cellular structure. Could the authors calculate the expansion factor by measuring cilia diameter in STED and expanded confocal samples?

We agree that a direct check in the sample structure is more sensitive to local change in expansion as well as isotropicity compared to the overall gel especially for the study that precise amount of expansion factor is important. However, to measure the diameter of the same cilia before and after expansion is almost impossible regarding the numerous and packed cilia in the spinal cord samples. In addition to calculating the expansion factor by measuring the size of the sample/gel before and after expansion, we measured the diameter of several cilia (before and after expansion) and calculated the expansion factor by using the means of the diameter from STED and ExSTED. We also made a graph for comparison of cilia diameters (divided by the expansion factor in the expanded one) in the different techniques (the graph added in the Figure 5—figure supplement 2).

Reviewer #2 (Recommendations for the authors):1. On the microscopy section, the authors make a statement that DSVs within somatostatin-positive sensory cells are present in the axons at baseline and become more numerous upon pH variations. To illustrate their point, the authors need to show :a) Images illustrating that DSVs were found in the axonal extension (page 5, "data not shown") at baseline, and;

Not visible in Figure 2 but in Figure 3 D-F.

b) Evidence determining whether DSVs are released or transported from the soma to the axon (Figure 3d-f) upon changes of pH. A quantification of the total number of DSV per cell could be insightful to resolve these 2 interpretations.This point is important for interpretation of the mechanisms at play upon variations of pH.

We quantified the number density of vesicles per soma area (µm^-2^) (2DSTED) for somatostatin and dopamine DCVs (explained above). We have additionally applied volumetric imaging and quantified the number density of somatostatin and dopamine DCVs within soma volumes (µm^-3^) added in Figure 3—figure supplement 1 and figure 4—figure supplement 1.

2. The interpretation of physiological results based on established knowledge of ASICs in lamprey cannot be easily explained, and the authors need to provide further data to support their claims or rephrase their conclusions:a) the authors need to show that somatostatin-positive cells respond to pH in their conditions as it is key for their interpretation, and their previous article on all spinal CSF-contacting neurons did not mention that only half the ciliated neurons contacting the CSF responded to pH (Jalalvand et al. Current Biology 2016).

The study of Jalalvand et al. 2016 dealt with one specific subtype of CSF-contacting cells that express somatostatin and are located at the lateral aspect of the central canal and has axonal projections to the lateral margin of the spinal cord (see also new Figure 8). It was known from previous studies that another subtype of CSF-contacting neurons in lamprey expressed dopamine (Schotland et al., 1996), in which the cells have a more ventral location. The latter with a previously unknown function has now been included in the present study and a function unravelled.

b) Similarly, the authors here examine the mechanosensory response of dopaminergic CSF-contacting neurons without quantifying the response of somatostatin-positive neurons to the same stimulus. Both responses to variations of pH and mechanical stimulation need to be performed in both cell types.

We have previously reported on the sensitivity of the somatostatin-CSF-contacting neurons (Jalalvand et al. 2016, Nature Communications) specifically showing that they respond to both fluid movements and pH changes, and that both responses are blocked by APETx2, a specific ASIC3 antagonist. Since these responses have been quantified in considerable detail earlier, we feel that there would be no reason to repeat these results in the context of these experiments.

c) Another group have shown that lamprey only express one type of ASIC channel, ASIC1 (not the channel ASIC3 ): see https://www.ncbi.nlm.nih.gov/pmc/articles/PMC3047259/). Therefore, how can the authors explain the effect of the drug used against ASIC3 in this context ?

We note that in the article of Coric et al. 2005, they had identified one clone of cDNA that corresponded to ASIC1 and when expressed in oocytes, no pH sensitivity was found. They do not comment regarding the possible presence of ASIC3. The review of Grunder and Chen (2010) has a focus on ASIC1a and base their comment on lamprey on Coric et al. 2005. Our evidence for the presence of ASIC 3 in lamprey is that both the mechanical and pH response are blocked by APETx2 (Jalalvand et al., 2016). ASIC3 is present in both the peripheral and central nervous system in mammals.

In order to solve this issue, the authors could show expression of ASIC channels, and test the effect of a drug against ASIC1 on the response of somatostatin-expressing cells.d) Finally, the only ASIC channel expressed in lamprey, ASIC1, when cloned and tested, has been reported to be proton insensitive (see https://www.ncbi.nlm.nih.gov/pmc/articles/PMC1464184/) – how then to explain then the activation of a subset of ciliated neurons exposed to protons?

It is correct (Coric et al. 2005) that the ASIC1 clone extracted from lamprey was not pH sensitive when expressed in frog oocytes, but our data show that the mechanical/pH response in the lamprey somatostatin expressing CSF cells in lamprey is mediated by ASIC3, and there is no evidence to suggest that ASIC3 should not be present in lamprey. The dopamine-containing cells are not pH-sensitive.

[Editors' note: further revisions were suggested prior to acceptance, as described below.]

[…] Specific comments from one reviewer:The authors satisfactorily answered some of the issues raised on the quantification of images and provided high resolution images for their outstanding data acquired at high resolution. These data are very convincing and beautiful. Congratulations, this will be very precious and inspirational for further studies in the spinal field !The issue remaining are related to the interpretation of the physiological results. The authors should either show expression of ASIC3 in CSF-cNs by ISH or tone down their conclusions relative to the ASIC3 receptor in response to pH.

We have reworded our text on page 8 and 13.

Similarly the authors should be more careful for stating the role PKD2L1 in pH sensing or mechanoreception as they lack genetic methods to investigate its function in lamprey.

Since the mechanosensitivity is mediated by an APETx2 sensitive mechanism in somatostatin CSF-c neurons but through another mechanism in the dopamine CSF-c neurons it is natural to point out the possibility of PKDL1, which is responsive in the zebrafish.

The text on page 8 now reads:

“As PKD2L1 has been confirmed as a mechanosensitive ion channel in zebrafish (Bohm et al., 2016, Sternberg et al., 2018, Orts-Del'Immagine et al., 2020), the results suggest that the mechano-sensitivity of the dopaminergic CSF-c neurons may be mediated by PKD2L1 channels.”

A) On the role of ASIC3 in pH sensing :As stated in my first review, ASIC3 can only be found in the genome of mammals. Only ASIC1 has been identified in the lamprey genome (https://www.ncbi.nlm.nih.gov/pmc/articles/PMC3047259) and the lamprey ASIC1a is proton insensitive (https://www.ncbi.nlm.nih.gov/pmc/articles/PMC1464184/).The authors did not respond to my request of testing the role of ASIC1a in their experiments using pharmacology.

We have not tested ASIC1a blockers, since this channel in lamprey is known to not respond to pH (Coric et al. 2005), and ASIC1a is not blocked by APETx2.

One can therefore wonder whether 2uM APETx2 used by the authors (about 100X the EC50 of ASIC3 homomers, https://pubmed-ncbi-nlm-nih-gov/17113616/) could be acting on other targets (see suggested actions on Na or K channels mentioned here, https://pubmed-ncbi-nlm-nih-gov/17113616/). Please discuss clearly in the text.

We work within the specific dose range as described by Diochot et al. 2004, and the same range as has been used in mammalian dorsal root ganglion cells. We also note that APETx2 blocks (the ASIC3 blocker) the mechanosensitive response in somatostatin cell, whereas it has no effect on the mechanosensitivity of dopamine CSF-c cells. This strongly suggests that the effect of APETx2 is not a general unspecific effect on lamprey neurons.

We have added (the following text in the Discussion on ine page 13):

“The mechano-sensitivity of the dopamine and somatostatin expressing CSF-c neurons are mediated by different cellular mechanisms, the latter is blocked by APETx2, a selective ASIC3 blocker as shown in previous studies (E. Jalalvand, B. Robertson, H. Tostivint, et al., 2016; E. Jalalvand, B. Robertson, P. Wallen, et al., 2016). et al. et al. […] The data supports the previous interpretation that ASIC3 mediates both the acid-sensing and the mechanosensitivity in these CSF-c neurons.”

We also responded previously to this issue in our response:

In the article of Coric et al. 2005 they had identified one clone of cDNA that corresponded to ASIC1 and when expressed in oocytes, no pH sensitivity was found under these conditions. They do not comment regarding the possible presence of ASIC3. The review of Grunder and Chen (2010) has a focus on ASIC1a and base their comment in lamprey on Coric et al. 2005. Our evidence for the presence of ASIC3 in lamprey is that both the mechanical and pH response are blocked by APETx2, a selective antagonist of ASIC3, (Jalalvand et al. 2016, Nature. Com), strongly suggesting the presence of ASIC3 in the lamprey. ASIC3 is present in both the peripheral and central nervous system in mammals.

B) On the role of PKD2L1 in mechanoreception:The authors summarize that PKD2L1 is responsible for mechanoreception only in dopaminergic CSF-cNs.However:1. PKD2L1 is expressed in all CSF-cNs, ventral (dopaminergic) and dorsal (somatostatinergic) as shown by the authors here and found in zebrafish (Djenoune 2014) and mouse (Huang 2006; Petracca 2018). In fact, in their 2016b publication, the authors had proposed that SST+ CSF-cNs were responding to basic pH via PKD2L1, suggesting a role for this channel in these cells.Why therefore in this study, only mentioning the role of PKD2L1 for mechanoreception in the dopaminergic CSF-cNs despite the expression being there in both dorsal and ventral cell types?

Our *results* clearly show that there are different cellular mechanisms for the mechanotransduction, one blocked by APETx2 in somatostatin CSF-c cells and not in dopamine CSF-c neurons. Since PKD2L1 is present in the dopamine cells and known to be mediate mechanosensitivity in zebrafish CSF-c cells it is natural to point to this possibility.

2. Due to the lack of specific antagonists, the authors do not have the tools in lamprey to measure its contribution to mechanoreception. They can only suggest that PKD2L1 contribute to mechanoreception in both ventral and dorsal CSF-cN types.

The mechanosensitive mechanisms in the two types of CSF-c cells, the laterally projecting and ventral CSF-c neurons, in lamprey must be different, given the difference in APETx2 effects.

3. Please correct an error in citation and references used:The authors use Bohm et al. 2016 to state that CSF-cNs rely on PKD2L1 to be mechanosensory in zebrafish.To be correct, we showed in vivo in Bohm 2016 that both ventral and dorsal CSF-cNs respond to concave (not convex) mechanical deformations of the spinal cord via PKD2L1 (response are abolished in the KO).However, mechanoreception cannot be rigorously demonstrated in vivo. It is therefore only in Sternberg 2018 that we could show in vitro using a piezo device to mechanically stimulate their membrane that all CSF-cNs isolated in primary cultures are mechanosensory cells and that their response always rely on PKD2L1.Note that in zebrafish, we now understand that CSF-cNs in vitro do not respond to CSF flow (Prendergast et al. under review) and in vivo, their response to concave mechanical bending of the spinal cord needs their interaction with the Reissner fiber (Orts Dell Immagine 2020), which does not alter itself the flow (Cantaut-Belarif 2018).

Interesting with the results with Reissner´s fiber – a possibility that has been considered often but without substantial evidence previously, but not citable as yet.

The authors should at minima cite Sternberg et al. 2018 for showing the role of PKD2L1 in mechanoreception, but to be fair, also propose with more nuances in the discussion how these cells in lamprey can combine ASIC and PKD channels to sense pH and mechanical inputs, citing Bohm 2016, Sternberg 2018 and Orts Del Immagine 2020.

Done.